# ICAM-1/CD18-mediated sequestration of parasitized phagocytes in cortical capillaries promotes neuronal colonization by *Toxoplasma gondii*

Matias E. Rodriguez ⓘ, Ali Hassan, Nikolaos Linaroudis, Felix Harryson-Oliveberg, Arne L. ten Hoeve ⓘ & Antonio Barragan ⓘ ✉

Microbial translocation across the blood-brain barrier (BBB) is a prerequisite for colonization of the central nervous system. The obligate intracellular parasite *Toxoplasma gondii* chronically infects the brain parenchyma of humans and animals, in a remarkably stealthy fashion. We investigated the mechanisms of BBB traversal by *T. gondii* (genotypes I, II, III) and *T. gondii*-infected leukocytes, using intracarotid arterial delivery into the cerebral circulation of mice. Unexpectedly, parasitized dendritic cells (DCs) and other peripheral blood mononuclear cells were found to persistently sequester within cortical capillaries. Post-replicative egress of *T. gondii* from sequestered DCs was followed by rapid parasite localization within cortical neurons. Infection-induced microvascular inflammation dramatically elevated the sequestration of parasitized DCs, while treatments targeting the ICAM-1/CD18 leukocyte adhesion axis with blocking antibodies strongly reverted sequestration. The parasite effectors TgWIP and GRA15, known to promote leukocyte hypermigration and inflammatory activation, further increased both the capillary sequestration of infected DCs and cerebral parasite loads in a strain-dependent manner. These findings reveal that the sequestration of parasitized leukocytes in cortical capillaries, with subsequent BBB traversal following parasite egress, provides a mechanism for *T. gondii*'s rapid access to cortical neurons during primary infection.

The blood-brain barrier (BBB) protects the vertebrate central nervous system (CNS) from most infectious insults, with maintained homeostasis and regulated passage of cells to the brain parenchyma[1,2]. Anatomically localized to the cerebral microvasculature, the BBB is constituted by arterioles, capillaries and venules[3,4]. Two internal carotid arteries (ICAs) and 2 vertebral arteries merge at the base of the brain in the circle of Willis to supply the brain with blood. From the circle of Willis, inter-cerebral and pial arteries lead the blood to arterioles, which penetrate the parenchyma and lead to the capillary network[5,6]. With an estimated total length of 600 km (370 miles) and a luminal surface area of 15–25 m² (50–82 sq. ft.) in the adult human, the BBB vasculature is dominated by capillaries of luminal diameter <10 μm, surrounded by a neurovascular unit (NVU) wall of <10 μm and an average cortical inter-capillary distance of 40 μm[7,8]. It is becoming increasingly clear that neuro-vascular interactions are crucial to maintain homeostasis in the CNS[9].

The microvascular lumen of the BBB is covered by highly specialized endothelial cells characterized by low permeability, low

Department of Molecular Biosciences, The Wenner-Gren Institute, Stockholm University, 10691 Stockholm, Sweden. ✉e-mail: antonio.barragan@su.se

pinocytic activity and high transcellular electrical resistance[10]. Intercellular tight junction proteins restrict the paracellular flux of hydrophilic molecules and regulates the migration of cells across the endothelial barrier[11]. Basal lamina, pericytes and astrocyte 'end-feet' also contribute to the restricted permeability of the NVU[12,13]. Immune activity by resident cells and infiltrating leukocytes, in the absence or presence of neuroinflammation[14], are crucial to eradicate pathogens that succeed in translocating across the restrictive CNS barriers[2]. However, the generated inflammation can also perturb BBB functions[1] and alter neuronal function[15].

CNS infections by microbial pathogens are among the most devastating infectious diseases worldwide, often with highly limited therapeutic options[16]. The Apicomplexan parasite *Toxoplasma gondii* infects a broad range of warm-blooded vertebrates, with an estimated 1/3 of the global human population being chronically infected[17]. Three clonal lineages of *T. gondii* (type I, II, III) predominate in humans and animals in Europe and North America[18]. From its point of entry in the intestine, *T. gondii* disseminates systemically via the blood circulation and rapidly achieves latent infection in the CNS[19]. While chronic infection is generally considered asymptomatic, reactivated or acute infection can lead to life-threatening encephalitis in immune-compromised individuals and to severe neurological disorders in neonates[20].

The rapidly replicating invasive stage of *T. gondii* -the tachyzoite- is obligate intracellular and uses gliding motility to actively invade cells where it replicates[21]. Gliding motility also facilitates transmigration of tachyzoites across polarized endothelium in vitro[22,23] and provides an effective mechanism of propulsion in the microenvironment inside tissues[24]. In their journey to form chronic cysts, the tachyzoites cross the cellular brain barriers to establish latent infection in the CNS[25]. *T. gondii* also exploits the trafficking of mononuclear phagocytes for dissemination. Systemic spread from the intestine to peripheral organs via the blood circulation is largely mediated by parasitized CD11c[+] / CD11b[+] leukocytes[26]. Dendritic cells (DCs) and other phagocytes can act as *Trojan horses* for systemic dissemination of *T. gondii* in mice[27], in a parasite genotype-related fashion[28]. Upon active invasion by *T. gondii*, DCs are induced to migrate via activation of non-canonical GABAergic signaling and MAP kinase activation[29–32]. This migratory activation, termed hypermigratory phenotype[33], implicates secreted parasite effectors[34–36] and chemotactic activation of parasitized phagocytes[37].

Thus, a number parasite dissemination pathways and mechanisms of passage to the brain have been proposed in recent years. Yet, the relative contribution or importance of the different putative pathways mediating passage across the BBB have remained unresolved[24,25]. Also, studies have been partly hampered by the low parasite numbers reaching the brain parenchyma early during infection, forcing characterizations at later time points with exacerbated systemic infection[38,39]. To overcome these limitations and gain novel insights in the early mechanisms of passage, we developed an experimental system with delivery of *T. gondii* and parasitized phagocytes directly into the cerebral circulation.

## Results

### Parasitized CD45[+] leukocytes sequester in cerebral capillaries

While it is well established that mononuclear phagocytes mediate systemic and tissue dissemination of *T. gondii*, their association to the appearance of tachyzoites in the brain parenchyma has remained unresolved. To address this, we first performed intraperitoneal (ip) infections with *T. gondii* and detected increased numbers of peripheral blood mononuclear cell (PBMC)-associated tachyzoites in blood over time (Fig. 1A, B, Fig. S1A). Interestingly, analyses of brain cortices detected CD45[+] leukocytes with replicating GFP[+] tachyzoite vacuoles contained within cortical capillaries (Fig. 1C). However, experimental approaches in mice with delivery of *T. gondii* via the tail vein or ip[38,39] entail entering the systemic circulation with entrapment of circulating parasites in peripheral organs, especially the lungs and liver

(Fig. S1B–E). The restrictiveness of the BBB makes early quantifications of parasite numbers reaching the cerebral circulation difficult prior to replication and expansion locally, compared with other organs[39]. To overcome this known limitation, *T. gondii*-infected CFSE- or CMTMR-labeled DCs were injected directly into the cerebral circulation via the ICA (Fig. 1D), and the localization of parasitized cells was assessed in the frontal cortex 16 h post-inoculation (hpi) (Fig. 1E). Interestingly, GFP[+]CMTMR[+], RFP[+]CFSE[+] or CFSE[+] CMTMR[+] cells with replicating type I, II or III tachyzoites, respectively, were readily detected in the vasculature (Fig. 1F, G; Movie S1) with luminal diameter <10 μm (Fig. 1H), preferentially in the vicinity of vascular branching points and similarly for *T. gondii* type I-, II- and III-infected DCs (Fig. 1I). In contrast, higher numbers of type II-infected DCs compared with type I- and III-infected DCs were found in cortex (Fig. 1J), consistent with elevated adhesion of type II-infected DCs to polarized endothelial monolayers (Fig. 1K). We conclude that parasitized CD45[+] leukocytes, including DCs, can sequester in cortical capillaries, with measurable variations between *T. gondii* strains from clonal lineages I, II and III.

### Sequestration of parasitized DCs in cortical capillaries is rapidly followed by parasite egress and translocation of tachyzoites across the BBB

To determine the immediate fate of *T. gondii* tachyzoites contained in sequestered DCs, the brains of mice injected in the ICA were extracted 4–48 hpi (Fig. 2A) and the localization and numbers of infected DCs and *T. gondii* foci were microscopically assessed in relation to the vasculature. By 4 hpi, tachyzoites (GFP[+]) were exclusively associated with DCs (CMTMR[+]) within capillaries (Fig. 2B). Importantly, tachyzoites were found extra-vascularly by 16 hpi (Fig. 2B; Movie S2), indicating egress from DCs and passage across the endothelium. Expectedly, localization and replication within endothelial cells[38,39] with appearance of DC membrane remnants (*ghosts*) also occurred by 28 hpi (Fig. 2B; Movie S2). By 48 hpi, larger extravascular foci with replicating tachyzoite vacuoles appeared (Fig. 2B). Interestingly, while absolute numbers of infected DCs in the microvasculature decreased over time (Fig. 2C), numbers of non-DC-associated tachyzoites conversely increased (Fig. 2D), yielding a dramatic change in the ratio between the two (Fig. 2E). In contrast, numbers of non-infected DCs remained stable for 28 h (Fig. 2F), with maintained ratios related to infected DCs (Fig. 2G, H). Further, DCs infected with non-replicating *T. gondii* (CPS) remained adhered to the microvasculature for up to 48 hpi, in absence of replication or egress from DCs and strongly contrasting with the appearance of invasive parasitic foci upon *T. gondii*-WT challenge (Fig. 2I, J; Fig. S2). Jointly, the data show that replicating *T. gondii* tachyzoites egress from infected DCs sequestered in cortical capillaries and either transmigrate across endothelium or invade endothelial cells.

### *T. gondii* tachyzoites egressing from sequestered DCs are rapidly retrieved in cortical neurons

To address the fate of *T. gondii* following egress from infected DCs, tissues were stained for the neuronal nuclear and perinuclear cytoplasmic marker NeuN. By 16–28 hpi, replicating tachyzoites were detected in NeuN[+] cells (Fig. 3A, B; Movies S3, 4), evidencing a rapid translocation to neurons. Tachyzoites were also detected in the vicinity of astrocytes (GFAP[+]), albeit in the absence of major replication vacuoles (Fig. S3A). Next, to address if egressed free tachyzoites possess the ability to directly transmigrate from the vascular lumen across the BBB, freshly egressed tachyzoites were injected in the ICA and their localizations were assessed at 1 or 16 hpi. Both single WT tachyzoites and non-replicating CPS tachyzoites were retrieved in extravascular locations at 1 hpi, showing the rapidity of the tissue invasion or extravasation processes and that replication was not imperatively necessary (Fig. 3C; Movie S5, 6). While the vast majority of tachyzoites were located intravascularly or associated with the endothelium, in

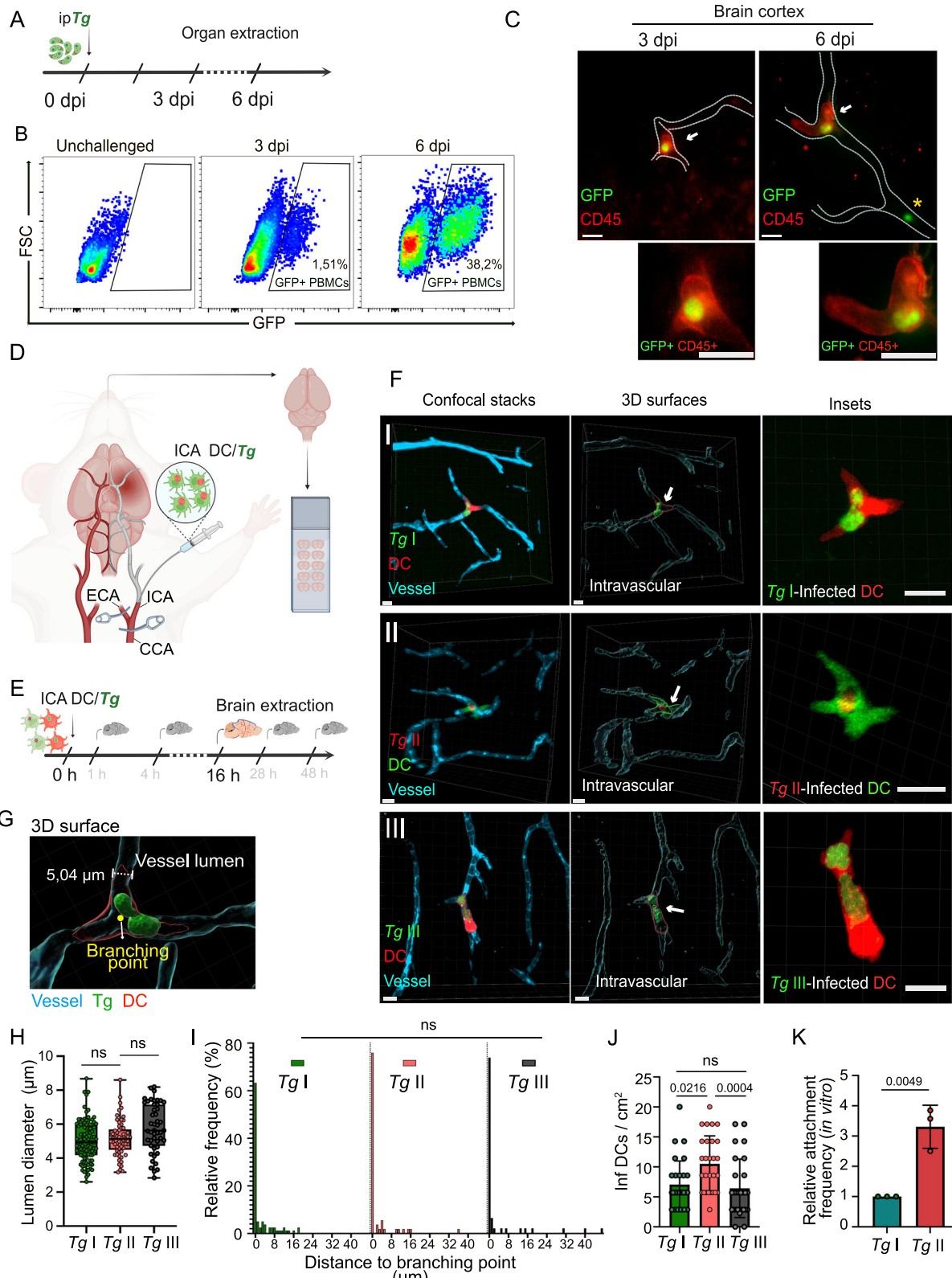

**A** ipTg — Organ extraction — 0 dpi / 3 dpi / 6 dpi

**B** Unchallenged / 3 dpi / 6 dpi — FSC vs GFP — 1,51% GFP+ PBMCs / 38,2% GFP+ PBMCs

**C** Brain cortex — 3 dpi / 6 dpi — GFP / CD45 — GFP+ CD45+

**D** ICA DC/Tg — ECA / ICA / CCA

**E** ICA DC/Tg — Brain extraction — 0 h / 1 h / 4 h / 16 h / 28 h / 48 h

**F** Confocal stacks / 3D surfaces / Insets
I — Tg I / DC / Vessel — Intravascular — Tg I-Infected DC
II — Tg II / DC / Vessel — Intravascular — Tg II-Infected DC
III — Tg III / DC / Vessel — Intravascular — Tg III-Infected DC

**G** 3D surface — Vessel lumen — 5,04 μm — Branching point — Vessel Tg DC

**H** Lumen diameter (μm) — Tg I / Tg II / Tg III — ns, ns

**I** Relative frequency (%) — Distance to branching point (μm) — Tg I / Tg II / Tg III — ns

**J** Inf DCs / cm² — Tg I / Tg II / Tg III — ns — 0.0216 / 0.0004

**K** Relative attachment frequency (in vitro) — Tg I / Tg II — 0.0049

line with previous observations[38,39], tachyzoite vacuoles were also identified at locations >10 μm from the capillary (Fig. 3D; Fig. S3B), implying extravascular localization in the parenchyma[7,8]. We also measured higher relative numbers of type I tachyzoites in cortical tissue compared with type II tachyzoites (Fig. 3E), with similar localization to capillaries (Fig. 3F; Fig. S3B) but longer distances to branching points in the vasculature (Fig. 3G). Typically, a 3–5 fold higher

inoculum dose of type II was needed to reach comparable numbers of parasite foci in the brain (Fig. S3C), likely reflecting a superior invasive capacity[24] and extracellular viability (Fig. S3D) by type I tachyzoites. In a similar fashion, we compared the relative tachyzoite numbers upon delivery of infected DCs or delivery as free tachyzoites, for both type I and type II strains. Consistent with microscopy analyses (Fig. 3E), inoculation of comparable cfu of free tachyzoites yielded higher

**Fig. 1 | Sequestration of infected PBMCs and DCs in the brain microvasculature.**
**A** Experimental set up. Freshly egressed GFP-expressing *T. gondii* (*Tg*) tachyzoites (PRU-GFP, $2 \times 10^5$ cfu) were inoculated ip in mice and organs were collected 3 or 6 days post-inoculation (dpi). **B** Representative flow cytometry plots of GFP vs forward scatter (FSC) show percentage of tachyzoite-associated PBMCs (GFP⁺) in unchallenged and infected mice 3 and 6 dpi (*n* = 3 mice). **C** Representative micrographs of brain cortex at 3 or 6 dpi. Mice were injected iv with α-CD45-Alexa 647 prior to organ extraction. Arrows indicate leukocyte (CD45⁺, red)-associated tachyzoites (GFP⁺, green). Dotted lines delimit cortical capillaries. Asterisk indicates tachyzoite (GFP⁺) non-associated to a leukocyte (CD45⁻). Lower panel insets show magnification of the leukocyte- associated tachyzoites. Scale bars: 10 μm.
**D** Cartoon shows microsurgical procedure, as detailed in Methods, with injection of cell suspensions (DC/*Tg*) into the internal carotid artery (ICA), followed by brain extraction and tissue sectioning (50 μm thickness). Common carotid artery (CCA) and external carotid artery (ECA) are indicated. Created in BioRender. Pairoto, M. (2025) https://BioRender.com/e26i498. **E** Experimental set up. CFSE or CMTMR pre-labeled DCs were challenged in vitro with *T. gondii* (RH-GFP, MOI 1; ME49-RFP, MOI 2; CFSE-prelabelled CTG, MOI 2) to obtain a DC infection frequency of ~50%. Infected DCs (-20 × 10⁶ DCs / - 10 × 10⁶ cfu *Tg*) were slowly (5 min) inoculated into the brain circulation via the ICA as detailed in Methods. Brains were extracted 16 h post-inoculation (hpi). **F** Confocal micrographs, with corresponding 3D surface

analyses, show the localization of *Tg* type I RH (GFP, green)-infected DCs (CMTMR, red), *Tg* type II ME49 (RFP, red)-infected DCs (CFSE, green) and *Tg* type III CTG (CFSE, green)-infected DCs (CMTMR, red) in relation to the vascular marker Evans blue (cyan). Arrows indicate infected DCs magnified in the insets, respectively. Scale bars: 10 μm. **G** 3D surface analysis as in (**F**) of microvessel (cyan) with infected DC (red) with *T. gondii* vacuoles (green). Microvessel lumen diameter (5,04 μm) and vascular branching point are indicated. **H, I** Luminal diameter (**H**) and distance to nearest vascular branching point (**I**) of cortical microvessels containing *T. gondii* type I, II, and III-infected DCs, respectively. In box plots, center line indicates median. Box limits: 25th and 75th percentiles. Whiskers: maximum and minimum values. Data are from 86 (*Tg*I), 61 (*Tg*II), 47 (*Tg*III) -infected DCs per condition from three independent experiments (*Tg*I *n* = 4 mice; *Tg*II, *Tg*III *n* = 3 mice). **J.** Bar graph shows mean (± SEM) numbers of *T. gondii* type I, II or III-infected DCs related to cortical area, from 30 (*Tg*I), 28 (*Tg*II) and 27 (*Tg*III) cortical sections per condition from 3 independent experiments (*n* = 3 mice per condition). **K** Bar graph shows the relative attachment frequency of *T. gondii* type I- or type II-infected DCs to polarized brain endothelial cells (bEnd.3) in vitro. Data are expressed as mean (± SEM) from three independent experiments (*n* = 3 biological replicates). Statistical analyses: (**H–J**) Kruskal–Wallis test followed Dunn's post-hoc test (**K**) 2-tailed unpaired Student's *t*-test, numeric *p*-values are indicated, ns: non-significant, *p* > 0,05. Source data are provided as a Source Data file.

parasite loads for type I (Fig. 3H). Further, while delivery in infected DCs yielded higher parasite loads for both type I and type II compared with free tachyzoites (Fig. 3I), type II loads exhibited a superior relative increase upon delivery in DCs, related to delivery as free type II tachyzoites (Fig. 3J). Jointly, the data align, and extend to the BBB, both the ascribed higher relative dependence on leukocytes for systemic dissemination by type II strains[28] and the higher transmigration frequencies in vitro by extracellular type I tachyzoites[24]. We conclude that tachyzoites egressing from sequestered DCs or delivered into the cerebral circulation as free tachyzoites can transmigrate across the BBB strain/type-dependently and in the absence of replication, thereby expediting the invasion of neurons.

### Sequestration of parasitized DCs depends on ICAM-1/CD18 and is exacerbated by a rapid systemic microvascular inflammation upon *T. gondii* infection

To determine the fate of *T. gondii* under inflammatory conditions in the CNS, infected DCs were injected in the ICA of animals pre-treated with pro-inflammatory LPS. Alternatively, heparin, which has anti-inflammatory and anti-adhesive effects, was administered (Fig. 4A). Pre-injection with LPS or heparin treatment dramatically elevated and reduced, respectively, the numbers of sequestered DCs in the vasculature (Fig. 4B, C). Interestingly, LPS-treatment elevated DC numbers in capillaries but not in post-capillary venules or arterioles (Fig. 4D, E), indicating selective effects. LPS and heparin, respectively, had similar opposite effects in vitro on endothelial monolayers, corroborating direct impacts on the adhesion frequencies of parasitized DCs (Fig. S4A). In vivo, LPS treatment elevated adhesion for both type I- and type II-infected DCs (Fig. S4B) and for *T. gondii*-challenged splenocytes (Fig. S4C–E), indicating that the pro-inflammatory effects were not highly dependent on parasite strain or host cell/leukocyte type. Next, we assessed if pre-infection with *T. gondii* ip also impacted numbers of sequestered DCs upon ICA injection. First, we confirmed that pre-infection ip yielded undetectable parasite loads in the brain at 48 hpi, yet quantifiable loads in lungs and liver (Fig. S4F–H). Interestingly, upon ICA injection, significantly higher numbers on sequestered infected DCs were detected in pre-infected animals (Fig. 4F–I), similar to the effects of pre-treatment with LPS. Of note, while the quantitatively predominant phenomenon was sequestration of parasitized DCs in the microvasculature, extravascular localization of infected DCs in the parenchyma was also demonstrated (Fig. S4I; Movie S7). However, the low frequency of this event precluded consistent quantification.

Next, we investigated the inflammatory response. A surprisingly rapid (<24 h) transcriptional upregulation of adhesion molecules,

including ICAM-1, and inflammatory markers in freshly-isolated cerebral microvessels from *T. gondii* ip-challenged mice (Fig. 4K; Fig. S4J). Reciprocally, parasitized DCs maintained CD18 expression (Fig. S4K). These findings, together with the paramount role of the ICAM-1/CD18 axis in leukocyte vascular adhesion, motivated treatments with blocking antibodies[40]. Importantly, both treatment with anti-ICAM-1[41,42] and anti-CD18[43], but not isotype control-treatments, dramatically reduced numbers of sequestered DCs (Fig. 4M, N; Fig. S4L) and parasite loads (Fig. 4O, P). We conclude that, upon *T. gondii* infection, a systemic inflammatory response rapidly activates the cerebral endothelium which facilitates sequestration of parasitized leukocytes in cortical capillaries, with an important implication of the ICAM-1/CD18 adhesion axis.

### Secreted parasite effectors impact the sequestration of infected DCs and cerebral parasite loads

Because *T. gondii* effectors TgWIP and GRA15 modulate the migratory features and inflammatory activation of parasitized leukocytes in vitro and in vivo[36,37,44], we assessed their putative impact on the sequestration of parasitized DCs in cerebral capillaries. Interestingly, mice challenged with Δ*gra15* or Δ*TgWIP*-infected DCs (Fig. 5A), consistently exhibited lower numbers of sequestered parasitized DCs in capillaries (Fig. 5B, C) and total parasite loads (Fig. 5D, E). Next, to determine the role of inflammation in this process, mice were pre-challenged with WT *T. gondii* ip and brains were extracted 1 hpi (Fig. 5F; Fig. S5A). Consistent with previous data (Fig. 4), total numbers of sequestered DCs increased dramatically under inflammatory condition, with maintained differences between WT and mutants (Fig. 5G–J; Fig. S5B), indicating direct effects on sequestration by TgWIP and GRA15. Consistently, Δ*gra15*-infected DCs exhibited reduced adhesion to endothelial cell monolayers in vitro, compared with WT-infected DCs (Fig. S5C). Similarly, a re-elevation of sequestration was measured upon challenge with a complemented mutant (Δ*gra15* + GRA15), reaching sequestration levels similar to WT (Fig. S5D). Next, we determined the long-term impact of GRA15 on parasite loads in the brain. Plaquing assays confirmed significantly lower cerebral parasite loads in mice challenged with Δ*gra15*-infected DCs at 7 dpi, with non-significant differences in the spleen (Fig. 5K, L). Further, we confirmed that pro-inflammatory treatment (LPS) and anti-inflammatory treatment (heparin) exacerbated and reduced, respectively, cerebral parasite loads at 7 dpi (Fig. 5M, N; Fig. S5E). Finally, we evaluated whether anti-ICAM-1 treatment affected cerebral parasite loads in mice at 7 dpi. Since pre-inoculation ip with wild-type *T. gondii* could contribute to total cerebral parasite loads at this stage, we instead used the non-replicative CPS strain. Prior to this, we confirmed that CPS induced an inflammatory

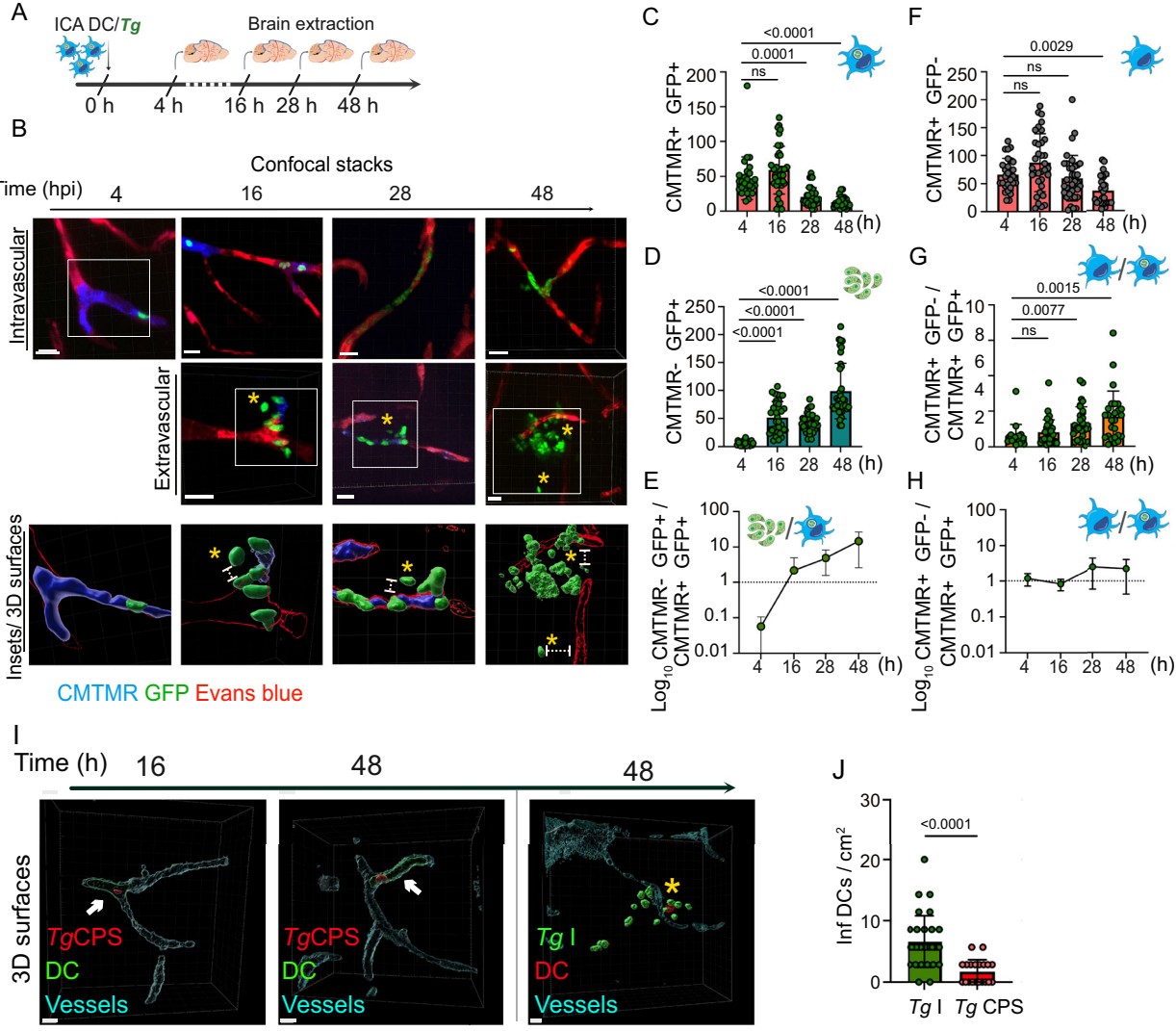

**Fig. 2 | Replication and egress of *T. gondii* from sequestrated DCs precedes parenchymal localization. A** Experimental set up. CMTMR pre-labeled DCs were challenged in vitro with *T. gondii* (RH-GFP, MOI 1) to obtain a DC infection frequency of ~50%. Infected DCs (20 × 10⁶ DCs/10 × 10⁶ cfu *Tg*) were inoculated in the brain circulation via the ICA and brains collected 4, 16, 28 or 48 hpi. **B** Confocal micrographs, with corresponding 3D surface analyses, show the localization of RH (GFP⁺, green)-infected DCs (CMTMR⁺, blue) in relation to the vascular marker Evans blue (red) at 4, 16, 28 and 48 hpi. White boxes (insets) show infected DCs (CMTMR⁺ GFP⁺) and non-DC associated *T. gondii* foci with replicating tachyzoites (CMTMR⁻ GFP⁺). Asterisks indicate extravascular non-DC associated *T. gondii*. Scale bars: 10 μm. **C, D** Graphs show the absolute numbers (mean ± SEM) of *T. gondii*-associated (infected) DCs (CMTMR⁺ GFP⁺) (**C**) and non-DCs associated *T. gondii* foci (CMTMR⁻ GFP⁺) (**D**), respectively, per cm² of cortical tissue. Data are from 24 (4 h), 36 (16 h), 37 (28 h) and 37 (48 h) cortical sections per timepoint from three independent experiments (*n* = 3 mice per condition). **E** Graph shows the Log₁₀ ratio between infected DCs (CMTMR⁺ GFP⁺) and non-DCs associated *T. gondii* (CMTMR⁻ GFP⁺) from (**C, D**). **F** Graphs show the absolute numbers (mean ± SEM) of non-

infected DCs (CMTMR⁺; GFP⁻) per cm² of cortical tissue. Data are from 28 (4 h), 34 (16 h), 36 (28 h) and 29 (48 h) cortical sections per timepoint from three independent experiments (*n* = 3 mice per condition). **G, H** Graphs show the mean (± SEM) ratio (**G**) and Log₁₀ ratio (**H**) between non-infected DCs (CMTMR⁺; GFP⁻) and infected DCs (CMTMR⁺; GFP⁺), respectively. Data are from 19 (4 h), 34 (16 h), 32 (28 h), 26 (48 h) cortical sections per timepoint from three independent experiments (*n* = 3 mice per condition). **I** Confocal 3D surface analyses show intravascularly located (Evans blue, cyan) RH-CPS infected DCs (CFSE⁺; mCherry⁺) at 16 and 48 hpi (left panels) and extravascular RH-WT tachyzoites (GFP⁺) at 48 hpi (right panel). Arrows indicate infected DCs. Asterisk indicates DC CFSE⁺ remnant (*ghost*, red). Scale bars: 10 μm. **J** Bar graph shows, for RH-WT and RH-CPS, the absolute numbers (mean ± SEM) of *T. gondii*-infected DCs retrieved in cortical sections at 16 hpi. Data are from 30 cortical sections from 3 independent experiments (*n* = 3 mice). **C–H** For each graph, cell and/or parasite cartoons indicate measured condition(s). **C–G** Kruskal–Wallis followed by Dunn's multiple comparisons test, (**J**) 2-tailed Mann–Whitney *U*-test, numeric *p*-values are indicated, ns: non-significant, *p* > 0,05. Source data are provided as a Source Data file.

response in the cerebral endothelium, similar to wild-type *T. gondii* (Fig. S5F, G). Infected DCs were then administered into the ICA ± anti-ICAM-1 treatment (Fig. 5O). Treated animals exhibited lower cerebral parasite loads compared to isotype control-treated mice, whereas splenic parasite loads remained unchanged (Fig. 5P). Altogether, we conclude that the *T. gondii* effectors TgWIP and GRA15 intensify ICAM-1-dependent sequestration of infected DCs in the brain vasculature, with an impact on cerebral parasite loads.

## Discussion

Stemming from the finding of sequestered parasitized CD45⁺ leukocytes in cortical capillaries, we undertook an experimental approach with controlled delivery of *T. gondii* into the cerebral circulation to assess early translocation events across the BBB. Strikingly, parasite replication within sequestered DCs was followed by egress and rapid intracellular localization in cortical neurons, with an impact on total cerebral parasite loads. This implied a sequential process with (*i*)

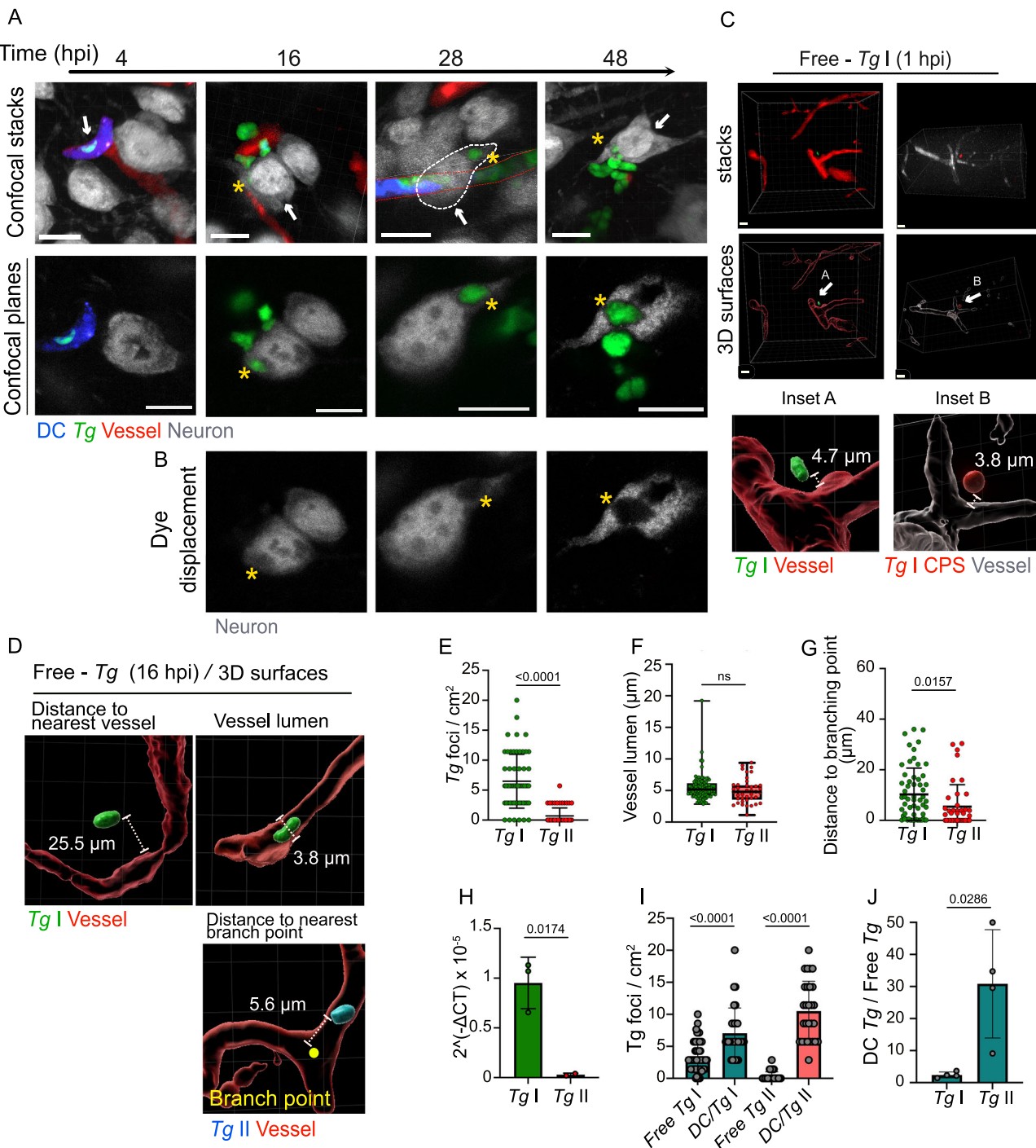

sequestration of parasitized leukocytes, (*ii*) niched intracellular replication in the sequestered leukocyte, (*iii*) egress and rapid extracellular transmigration across the BBB and (*iv*) subsequent infection of neurons (Fig. 6). We propose that this mechanism maximizes the transmigration efficiency and, consequently, facilitates the passage of *T. gondii* to the brain parenchyma. Indeed, it is upon egress from host cells that *T. gondii* tachyzoites peak in gliding motility and invasiveness, while these abilities decay over time[24,39].

The data establish that a persevering sequestration of parasitized leukocytes takes place in the cerebral microvasculature during *T. gondii* infection. Further, the sequestration of parasitized DCs was dramatically exacerbated by the inflammatory response to infection and by secreted parasitic effectors. Indeed, pre-infection in the peritoneum or LPS-treatment ip very rapidly activated the cerebral

endothelium, elevating transcriptional expression of cell adhesion molecules, such as ICAM-1, in microvasculature. Importantly, targeting ICAM-1 or its integrin counterpart CD18 with blocking antibodies[41–43], dramatically reduced sequestration and parasite loads, identifying a prime role for these two ligands in the sequestration of *T. gondii* within cortical capillaries. ICAM-1 came into focus due to its implication in transmigration of parasitized leukocytes in vitro[45,46] and also because it mediates adhesion and transmigration of free tachyzoites in vitro[23]. Along these lines, in cerebral *Plasmodium falciparum* malaria, ICAM-1 has been implicated in the sequestration of parasitized erythrocytes in the vasculature via parasite-derived variant adhesins[47]. Thus, the data indicates that the rapidly enhanced expression of ICAM-1 in cerebral endothelium upon *T. gondii* challenge facilitates adhesion of parasitized leukocytes first, and, later, transmigration of egressing

**Fig. 3 | Following sequestration of infected DCs, *T. gondii* is retrieved inside neurons. A.** Confocal micrographs show the localization of GFP-expressing RH tachyzoites (GFP⁺, green), DCs (CMTMR⁺, blue) and neurons (NeuN⁺, gray) in relation to the vascular marker Evans blue (red). Infected CMTMR pre-labeled DCs (20 × 10⁶ DCs / 10 × 10⁶ cfu *Tg*) were inoculated via the ICA and brains extracted at indicated time points. Upper panels show 3D projections from confocal stacks and lower panels show confocal planes. Arrows indicate infected NeuN⁺ cells. White and red dotted lines delineate neuronal cytoplasm and cortical capillary, respectively. Asterisks indicate GFP⁺ tachyzoites localized inside NeuN⁺ cells. Scale bars: 10 μm. **B** Confocal planes show displacement of NeuN signal (asterisks) by *T. gondii* tachyzoites in corresponding confocal planes in (**A**). **C** Confocal micrographs and corresponding 3D surfaces show the extravascular localization of RH wild type (GFP⁺, green) and non-replicative RH-CPS tachyzoites (mCherry⁺, red) in relation to the vascular marker Evans blue (red in left panels and white in right panels). 20 × 10⁶ cfu of freshly egressed tachyzoites were inoculated via the ICA and brains were extracted 1 hpi. Arrows indicate extravascular tachyzoites. Magnifications are shown in insets **A** and **B** for RH and RH-CPS, respectively, with distances the nearest blood vessel. Scale bars: 10 μm. **D** 3D surface analyses illustrate extravascular and intravascular localization of tachyzoites (type I RH, GFP⁺, green and type II ME49, RFP⁺, cyan) in relation to the vascular marker Evans blue (red). 20 × 10⁶ cfu of freshly egressed tachyzoites were inoculated via the ICA and brains were collected 16 hpi. Distance to the nearest vessel, vessel lumen and distance to the nearest branching point (yellow dot) are shown. **E** Graph shows the absolute numbers (mean ± SEM) of type I and II *T. gondii* foci per cm² brain tissue at 16 hpi. Data are from 59 (*Tg*I) and 56 (*Tg*II) cortical sections per condition from three independent experiments (*n* = 4 mice per condition). **F** Luminal diameter of cortical microvessels containing *T.*

*gondii* type I and type II tachyzoites, respectively. In box plots, center line indicates median. Box limits: 25th and 75th percentiles. Whiskers: maximum and minimum values. Data are from 89 (*Tg*I) and 33 (*Tg*II) *T. gondii* foci in cortical tissue per condition from three independent experiments (*Tg*I: *n* = 4 mice; *Tg*II *n* = 3 mice). **G** Distribution of distances to nearest vascular branching point (medians with 25th and 75th percentiles) of type I and II *T. gondii* retrieved in the cortical microvasculature. Data are from 59 (*Tg*I) and 39 (*Tg*II) *T. gondii* foci per condition from three independent experiments (*n* = 3 mice per condition). **H** Relative expression (qPCR) of *T. gondii* TgB1 gene at 16 hpi in cortices of mice challenged 20 × 10⁶ freshly egressed type I or type II tachyzoites inoculated via the ICA. Data are expressed as mean ( ± SEM) from three independent experiments (*n* = 3 mice per condition). **I** Graphs show the absolute numbers (mean ± SEM) of *T. gondii* foci or infected DCs per cm² brain tissue at 16 hpi. Mice were inoculated via the ICA with 10 × 10⁶ cfu of freshly egressed tachyzoites of type I RH (Free *Tg*I), type II ME49 (Free *Tg*II) or 10 × 10⁶ cfu of DCs pre-challenged with type I RH (DC/*Tg*I) or type II ME49 (DC/*Tg*II). Data are from 60 (free *Tg*I), 30 (DC/*Tg*I), 55 (free *Tg*II) and 28 (DC/*Tg*II) cortical sections per condition from three independent experiments (*Tg*I: *n* = 4 mice; *Tg*II *n* = 3 mice). **J** Ratios (mean ± SEM) between the relative numbers of infected DCs (from mice inoculated with DC/*Tg*) and *T. gondii* foci (from mice inoculated with Free *Tg*) retrieved in cortices for infections with RH (*Tg*I) and ME49 (*Tg*II), respectively. Data are from three independent experiments (*n* = 4 mice per condition). **E–J** 2-tailed Mann–Whitney U-test, (**H**) 2-tailed unpaired Student's *t*-test, **I** Kruskal–Wallis followed Dunn's multiple comparison test, numeric *p*-values are indicated, ns: non-significant, *p* > 0,05. Source data are provided as a Source Data file.

tachyzoites. Importantly, de-sequestration treatment by ICAM-1 blockade had an immediate (1- 24 h) and longer term (7 days) impact on cerebral parasite loads.

We demonstrate that two *T. gondii* effectors modulate the sequestration of parasitized DCs in cortical capillaries. Specifically, GRA15 elevated leukocyte sequestration in the microvasculature thereby facilitating parasite transmigration upon egress. This relative abundant and persistent sequestration, in the absence of a measurable passage of infected DCs to the parenchyma, is partly contrasting with the recent implication of both GRA15 and ICAM-1 in elevated transmigration of phagocytes across polarized endothelium in vitro[44,45]. The data also indicate that the adhesion to the vasculature by infected leukocytes does not automatically imply transmigration and underscore the importance of cautious extrapolations of in vitro phenotypes to conditions in vivo[48]. One possibility is that, very early during infection, parenchymal inflammatory cues that may promote leukocyte extravasation are not present. Further, the elevated sequestration by GRA15-expressing type II *T. gondii* is also consistent with the relatively higher dependency of type II strains on parasitized leukocytes for systemic dissemination[28] and the inferior sequestration by type I strain RH, which lacks a functional GRA15[49]. Also, because GRA15 activates NF-kB[50] and endothelial NF-kB mediates neuronal activation via TLR/Myd88 signaling[51], it is plausible that GRA15-mediated pro-inflammatory NF-kB activation facilitates colonization[52]. Further, TgWIP elevated sequestration and cerebral parasite loads, in line with its attributed roles in hypermotility and adhesion of DCs[36,44,53]. Jointly, because the type I strain RH expresses a non-functional form of GRA15[49], TgWIP deficiency in type I RH may phenotypically reflect a combined GRA15/TgWIP deficiency in this respect. This likely explains the overall inferior sequestration frequency of ΔTgWIP-infected DCs compared with ΔGRA15-infected DCs at very early time points (1 hpi), that is, before parasite egress or significant replication sets in. Finally, the recent discovery that a set of *T. gondii* effectors, including GRA15, induce CCR7-mediated chemotaxis of parasitized phagocytes elicits speculation[37,52]. The BBB endothelium constitutively expresses the chemokine CCL19[14] and, its receptor CCR7 has been suggested to mediate the migration of circulating T cells across the BBB[54]. However, direct evidence for endothelial CCL19 in mediating immune cell trafficking to the CNS in the absence of neuroinflammation is lacking and

awaits further investigations. Altogether, the findings highlight that the diversity and redundancy of polymorphic effectors, including co-operative effects[51], need to be taken into account when studying phenotypes linked to dissemination and colonization of the CNS by *T. gondii*. The data also gives rise to the intriguing possibility that type II tachyzoites compensate their inferior invasive capacity[24] of cerebral endothelium with a superior sequestration of infected leukocytes in the microvasculature.

We report that cortical neurons can contain replicating *T. gondii* as early as 16–28 h after intracarotid inoculation. Rapidly, tachyzoites were detected within the parenchyma, preferentially, but not exclusively, associated with neurons located in the immediate vicinity of the vasculature. The BBB (NVU wall) is <10 μm thick, with an average cortical inter-capillary distance of 40 μm[7,8] and an average distance of 15 μm between neuronal somata and capillaries[55]. Taking into consideration the ability of freshly egressed tachyzoites to transmigrate in vitro and migrate ex vivo in tissues[24,23,22], the present results extend in vivo the ability of *T. gondii* to translocate across non-permissive biological barriers, specifically the BBB. We confirm previous data of infection of endothelial cell linings[38,39] and add that endothelial cells and neurons can become infected simultaneously in time, indicating that replication within endothelium is common but not imperative for brain colonization. However, while circulating leukocytes facilitate systemic dissemination of *T. gondii*, it remains unclear whether free/extracellular tachyzoites in the blood contribute to dissemination, especially early during infection with low parasitemia, and given their sensitivity to neutralization by the complement system and IgM[56]. Here, we also provide evidence that extracellular tachyzoites and parasitized DCs, when delivered in the cerebral circulation, can directly enter the brain parenchyma, albeit at lower frequency, compared with tachyzoites entering after egress from sequestered DCs. Moreover, we extend previous findings that extracellular tachyzoites of type I (RH) transmigrate at higher frequencies than type II strains in vitro[24,39] to the BBB in vivo. Inherent differences between strains in gliding motility, invasiveness, and extracellular viability of tachyzoites likely contribute to these phenotypes[24]. The data indicate that direct transmigration across the BBB by circulating extracellular tachyzoites and direct transmigration by parasitized leukocytes represent relatively minor, but contributing, pathways to the initial colonization of the CNS.

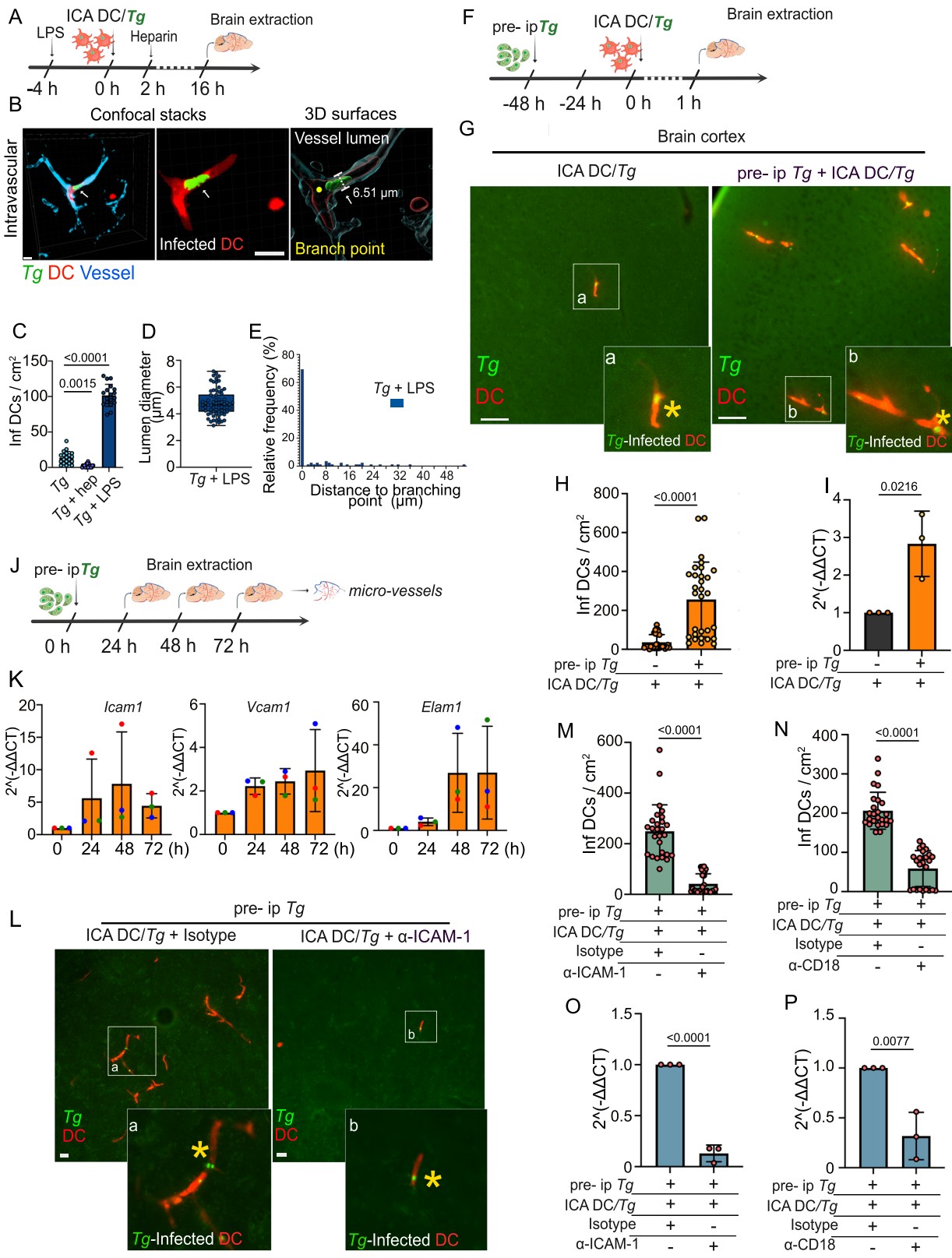

The preferential localization of *T. gondii* in cortical capillaries compared with other vascular segments after intracarotid inoculation is consistent with findings in infections ip or iv[38,39]. This localization contrasts somewhat with observations in human and experimental cerebral malaria. Malaria-parasitized erythrocytes bind preferentially to postcapillary venules and larger venules, with components of inflammatory leukocytes[57,58], but also to capillaries[59]. Further, the

preferential localization of *T. gondii*-infected DCs to vascular branching points is intriguing and reminiscent of the arrest of metastatic cancer cells in branching points of brain capillaries[60]. We report a surprisingly rapid (24 h) inflammatory activation of the cerebral microvasculature after intraperitoneal *T. gondii* infection, preceding the dramatically elevated systemic cytokine responses later during infection[61] and in line with the recently reported early inflammation of

**Fig. 4 | *T. gondii*-induced BBB inflammation promotes the sequestration of parasitized DCs. A** Experimental set ups. Infected CMTMR pre-labeled DCs ($20 \times 10^6$ DCs / $10 \times 10^6$ cfu *Tg*, PRU-GFP) were inoculated via the ICA and brains collected 16 hpi. When indicated, mice were pre-treated with LPS ip or treated with heparin iv. **B** Confocal micrographs and corresponding 3D surfaces show *T. gondii* (PRU-GFP)-infected DCs (CMTMR$^+$ GFP$^+$, red/green) in the cortical vasculature (Evans blue, cyan) of LPS pre-treated animals. Scale bars: 10 μm. **C** Graph shows the absolute numbers (mean ± SEM) of infected DCs per cm$^2$ of cortical tissue in control (*Tg*), heparin and LPS pre-treated conditions. Data are from 20 cortical sections per condition from three independent experiments (n = 3 mice per condition). **D, E** Mean vascular luminal diameter (**D**) and distance to nearest vascular branching point (**E**), respectively, of infected DCs retrieved in the cortical microvasculature in LPS pre-treated mice. In box plot, center line indicates median. Box limits: 25th and 75th percentiles. Whiskers: maximum and minimum values. Data are from 70 infected DCs from 20 cortical sections from three independent experiments (n = 3 mice). **F** Experimental set up. Mice were inoculated ip with $2 \times 10^5$ cfu of freshly egressed tachyzoites (ME49-RFP), (pre- ip*Tg*) or control medium. After 48 h, infected CMTMR pre-labeled DCs ($10 \times 10^6$ DCs / $5 \times 10^6$ cfu *Tg*, PRU-GFP) were inoculated via the ICA and brains collected 1 hpi. **G** Representative micrographs show *T. gondii* (PRU-GFP)-infected DCs (CMTMR$^+$, GFP$^+$) in cortical sections of control and pre-ip infected mice, respectively. Insets (a, b) show magnifications of white boxes and asterisks indicate infected DCs (CMTMR$^+$ GFP$^+$, red/green). Scale bars: 50 μm. **H, I** Graphs show the absolute numbers (mean ± SEM) of *T. gondii* (PRU-GFP)-infected DCs per cm$^2$ brain tissue at 1 hpi (**H**) and the relative expression

(qPCR) of *TgB1* gene in brain tissue (**I**), respectively. Data are from 30 cortical sections from three independent experiments (n = 3 mice). **J** Experimental set up. Freshly egressed RFP-expressing *T. gondii* (*Tg*) tachyzoites (ME49-RFP, $2 \times 10^5$ cfu) or control medium were inoculated ip in mice, brains were extracted 24, 48 and 72 hpi and micro-vessels purified. **K** Relative mRNA expression (qPCR) of *Icam1* (ICAM-1), *Vcam1* (VCAM-1) and *Elam1* (E-selectin) in brain micro-vessels at indicated time points. Data are expressed as mean (± SEM) from three independent experiments (n = 3 mice). **L** Representative micrographs show infected DCs (CMTMR$^+$ GFP$^+$) in the brain cortex of mice inoculated in the ICA with $10 \times 10^6$ DCs / $5 \times 10^6$ cfu *Tg* (PRU-GFP) plus α-ICAM-1 antibody or isotype. Mice were pre-infected ip with $2 \times 10^5$ cfu type II ME49-RFP tachyzoites. Insets (a, b) show magnifications of the infected DCs (CMTMR$^+$, GFP$^+$) and asterisks show tachyzoites. Scale bars: 20 μm. **M, N** Graphs show the absolute numbers (mean ± SEM) of *T. gondii* (PRU-GFP)-infected DCs per cm$^2$ brain tissue at 1 hpi in mice following inoculation with DC/*Tg* plus α-ICAM-1 (**M**) or α-CD18 (**N**), respectively. Mice were pre-infected (pre- ip, ME49-RFP). Data are from 30 cortical sections per condition from three independent experiments (n = 3 mice). **O, P** Relative expression (qPCR) of *T. gondii TgB1* gene in brain tissue of mice pre-ip (ME49-RFP) infected and inoculated in the ICA with infected CMTMR pre-labeled DCs ($10 \times 10^6$ DCs / $5 \times 10^6$ cfu *Tg*, PRU-GFP) plus α-ICAM-1 (**O**) or α-CD18 (**P**), respectively. Data are expressed as mean (± SEM) from three independent experiments (n = 3 mice). **C** Kruskal–Wallis followed Dunn's multiple comparison test, (**H**–**N**) 2-tailed Mann–Whitney *U*-test, (**I**–**P**) 2-tailed unpaired Student's *t*-test, numeric *p*-values are indicated, ns: non-significant, *p* > 0,05. Source data are provided as a Source Data file.

choroid plexus endothelium by day 3 post-infection[62]. In contrast, despite the very early neuronal infection, no evidence of inflammatory leukocyte infiltration, hemorrhage or disruption of the BBB[63] was found at these early timepoints, though transient focal elevations of BBB permeability[39] cannot be fully excluded. Importantly, these findings match well with the stealthy CNS colonization during primary *T. gondii* infection, which is typically clinically silent or accompanied by mild symptomatology in the absence of focal neurologic signs[20]. In sharp contrast, vascular colonization during meningococcal meningitis includes endothelial alterations, inflammation and purpuric lesions, and paracellular translocation of *Neisseria meningitidis*, although the fate of internalized bacteria has not yet been elucidated[2].

On a technical note, unilateral occlusion of the ICA can unlikely explain the observed sequestration pattern because circulation is rapidly compensated by collateral vessels and redistribution of blood at the circle of Willis[5,6], confirmed by the sequestration of parasitized DCs in the contralateral cerebral hemisphere and by a lack of focal symptomatology in mice post-procedure. Regardless of inoculation route, the colonization of the brain parenchyma by *T. gondii* consists of low-frequency events, related to the total parenchymal infiltration in other organs[39]. The undertaken approach allowed spatiotemporal quantitative analyses of extensive cortical areas at high resolution to measure low-frequency early events, which is limited using other approaches. The immediate effects of blocking antibodies avoided the adaptation and compensatory mechanisms in mutant mice owing to partly overlapping or redundant functions of CAMs and integrins[64,65].

In recent years, work in the field has described dissemination pathways for *T. gondii* and possible routes for CNS colonization[29,30,35,36,38,39,66–68]. Here, we add that the sequestration of parasitized leukocytes within cortical capillaries represents a remarkable mechanism enabling *T. gondii* translocation to the cerebral parenchyma and subsequent neuronal infection. In our experimental system, direct migration of *T. gondii*-infected DCs into the parenchyma was rare, occurring far less frequently than the CD18/ICAM-1-dependent sequestration of DCs. Thus, sequestered leukocytes, alike endothelium[38,39], constituted a replicative niche that facilitated direct transmigration of *T. gondii* tachyzoites upon egress. To better understand the dissemination of *T. gondii*, we argue that it is essential to distinguish between (*i*) the role of leukocytes in facilitating systemic dissemination to peripheral organs, including their transport to the

brain organ[26,27,66], and (*ii*) the mechanisms that mediate the pathogen's crossing of the BBB into the brain parenchyma itself[19,48]. The findings presented here establish a crucial pathogenetic link between systemic dissemination and CNS invasion in toxoplasmosis. Specifically, the data highlight how leukocyte sequestration acts as a bridge between these two critical stages of infection, ultimately facilitating neuronal colonization.

Importantly, cerebral parasite loads as a consequence of sequestration outnumbered loads after inoculation with free tachyzoites for type I and II strains. It is reasonable to speculate that a rapid invasion of neurons minimizes the risk of elimination by the immune system, permits a swift initiation of bradyzoite cyst formation and avoids excessive host-detrimental inflammation that lytic processes at the BBB entail. The ensuing immune response, which follows the initial parasite invasion to the parenchyma, likely neutralizes excessive replicating tachyzoites around vasculature and a few cysts can develop in neurons. We propose therefore that the adhesion of parasitized leukocytes to the cerebral vasculature provides, from the pathogen's perspective, a safe and cost-effective mechanism for passage to the CNS. From an evolutionary perspective, this relatively silent process may favor chronic asymptomatic carriage in the CNS of intermediate hosts, to ultimately ensure transmission to the feline definitive host upon predation. Future prophylactic or therapeutic approaches targeting cerebral toxoplasmosis need to take into consideration the diverse mechanisms used by *T. gondii* for colonizing the CNS.

## Methods
### Ethical considerations
The Regional Animal Research Ethical Board, Stockholm, Sweden, approved experimental procedures in mice and protocols involving extraction of cells from mice (permit number 16403-2022), following proceedings described in EU legislation (Council Directive 2010/63/EU). All methods were carried out in accordance with relevant guidelines and regulations. All methods are reported in accordance with ARRIVE (Animal Research: Reporting of In Vivo Experiments) guidelines (https://arriveguidelines.org).

### Mice
All experiments were performed using male and female C57BL/6NCrl mice (strain code 027, Charles River), aged 4 to 10 weeks, housed in a ventilated facility, provided with unrestricted access to tap water and

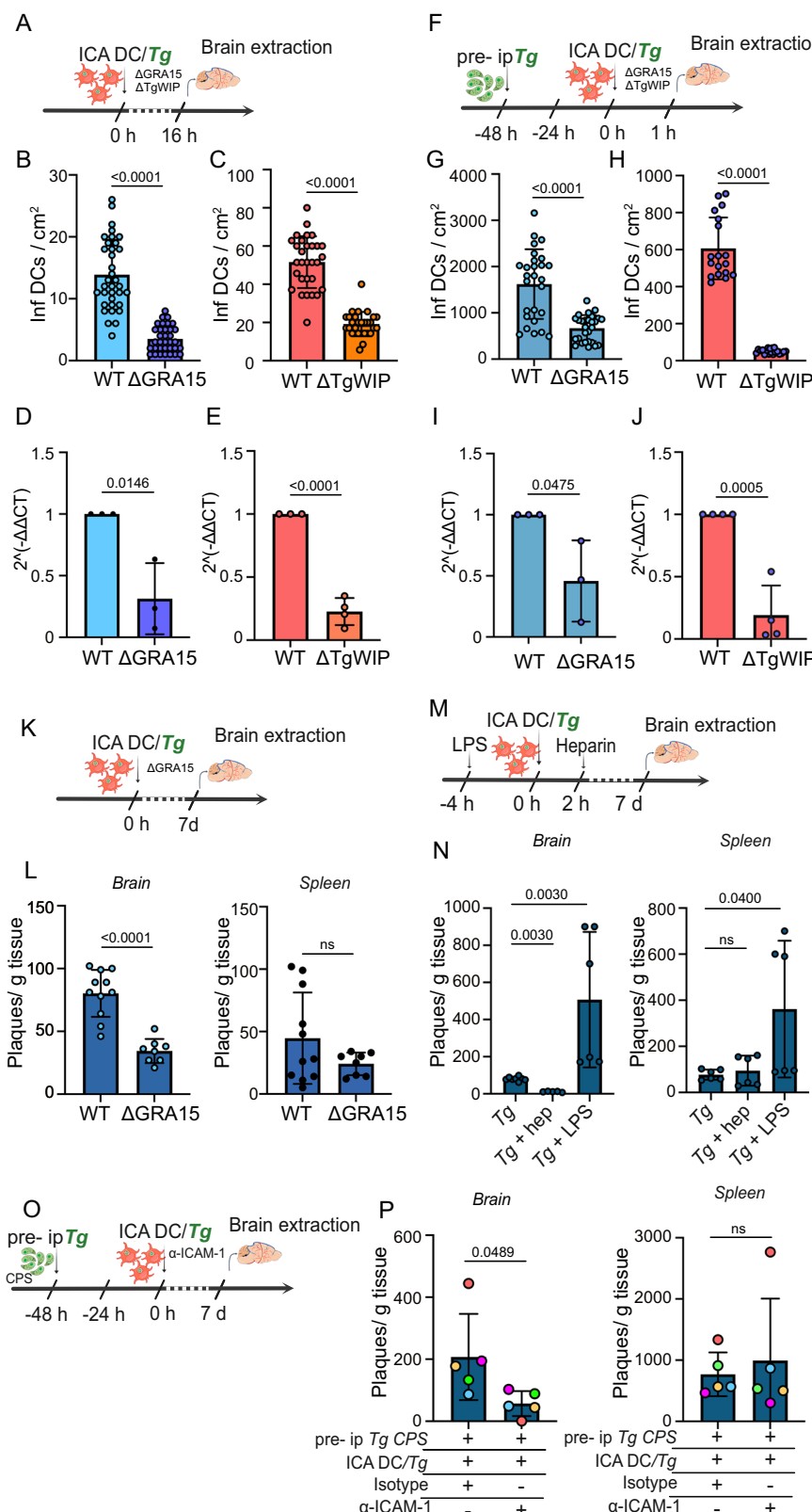

food, and kept under a 12-h light/dark cycle at a temperature of 20–22 °C.

## Parasite culture and cell lines

*T. gondii* tachyzoites of type I: RH-GFP, RHΔ*TgWIP*-GFP[36], RH CPS-mCherry[69], type II: ME49-RFP[70], PRU-GFP[50], PRUΔ*TgGRA15*-GFP[50], PRUΔ*TgGRA15* + GRA15-GFP[50] and type III: CTG (ATCC, 50842™) were

routinely maintained by serial 48 h passaging in human foreskin fibroblasts, HFFs (ATCC, CRL-2088). HFFs and murine brain endothelial cells, bEnd.3 (ATCC, CRL-2299), were cultured in High glucose Dulbecco's modified Eagle's medium (DMEM, VWR, Cat# 392-0415) supplemented with 10% heat inactivated fetal bovine serum (FBS, HyClon, Cat# SV30160.03), 20 µg/ml gentamicin (Gibco, Cat# 15710-049) and 10 mM HEPES (HyClone, Cat# SH30237.01). Cell cultures and

**Fig. 5 | *T. gondii* effectors TgWIP and GRA15 modulate the sequestration of parasitized DCs. A**. Experimental set up. CMTMR pre-labeled DCs were challenged with GFP-expressing wild type (*WT*), Δ*GRA15* or Δ*TgWIP* tachyzoites, followed by inoculation into the brain circulation via ICA (20 × 10⁶ DCs / 10 × 10⁶ cfu *Tg*). Brains were collected 16 hpi. **B, C** Graphs show the absolute numbers (mean ± SEM) of infected DCs (CMTMR⁺, GFP⁺) per cm² cortical tissue for PRU-*WT versus* PRUΔ*GRA15* (**B**) and RH-*WT versus* RHΔ*TgWIP* (**C**), respectively. Data are from 30 cortical sections per condition from three independent experiments (*n* = 3 mice). **D, E** Relative expression (qPCR) of *T. gondii TgB1* gene in brain tissue of mice inoculated with DC/*Tg* PRU-*WT versus* PRU Δ*GRA15* (**D**) and RH-*WT versus* RH Δ*TgWIP*, respectively. Data are expressed as mean (± SEM) from three independent experiments (*n* = 3 mice). **F** Experimental set up. Mice were inoculated ip with 2 × 10⁵ cfu of freshly egressed tachyzoites (ME49-RFP, pre- ip*Tg*). After 48 h, infected CMTMR pre-labeled DCs (10 × 10⁶ DCs / 5 × 10⁶ cfu *Tg*, *WT*, Δ*GRA15* or Δ*TgWIP*) were inoculated via the ICA and brains collected 1 hpi. **G, H** Graphs show the absolute numbers (mean ± SEM) of infected DCs (CMTMR⁺, GFP⁺) per cm² cortical tissue for PRU-*WT versus* PRUΔ*GRA15* (**G**) and RH-*WT versus* RHΔ*TgWIP* (**H**), respectively. Data are from 30 cortical sections per condition from three independent experiments (*n* = 3 mice). **I, J** Relative expression (qPCR) of *T. gondii TgB1* gene in brain tissue of pre- ip infected mice inoculated with DC/*Tg* PRU-*WT versus* PRUΔ*GRA15* (**I**) and RH-*WT versus* RHΔ*TgWIP* (**J**). Data are expressed as mean (± SEM) from three independent experiments (*n* = 3 mice). **K** Experimental set up. Infected CMTMR pre-labeled DCs (2 × 10⁴ DCs / 1 × 10⁴ cfu *Tg*, *WT* or Δ*GRA15*) were inoculated via the ICA and brains collected 7 dpi. **L**. Relative cerebral parasite loads in brain and spleen, respectively, upon challenge with PRU-*WT* and PRUΔ*GRA15*, determined by plaquing assays. Data show number (mean ± SEM) of plaques per gram of tissue from four independent experiments (*n* = 4 mice). **M** Experimental set ups. Infected CMTMR pre-labeled DCs (2 × 10⁴ DCs / 1 × 10⁴ cfu *Tg*, PRU-*WT*) were inoculated via the ICA and brains collected 7 dpi. When indicated, mice were pre-treated with LPS ip or treated with heparin iv. **N** Relative cerebral parasite loads in brain and spleen, respectively, of mice challenged with PRU-*WT* and pre/treated with LPS or heparin, determined by plaquing assays. Data show number (mean ± SEM) of plaques per gram of tissue from three independent experiments (*n* = 4 mice). **O** Experimental set up. Mice were inoculated ip with 10 × 10⁶ cfu of freshly egressed non-replicative tachyzoites, (RH-CPS-mCherry), (pre-ip *Tg* CPS). After 48 h, infected DCs (10 × 10⁴ DCs / 5 × 10⁴ cfu *Tg*, PRU-*WT*) plus α-ICAM-1 or isotype control were inoculated via the ICA. Brains were collected 7 dpi. **P** Cerebral parasite loads in brain and spleen, respectively, of mice challenged with DC/*Tg* PRU-*WT* plus α-ICAM-1 (or isotype), determined by plaquing assays. Mice were pre-infected (pre- ip, RH-CPS mCherry). Data show number (mean ± SEM) of plaques per gram of tissue from five independent experiments. For each experiment, mice were challenged and treated pairwise (α-ICAM-1 or isotype control) and datapoints are color-coded accordingly (*n* = 5 mice). **B–H** 2-tailed Mann–Whitney *U*-test, (**D–P**) 2-tailed unpaired Student's *t*-test, (**N**) One-way ANOVA followed by Bonferroni's multiple comparison test, numeric *p*-values are indicated, ns: non-significant, *p* > 0,05. Source data are provided as a Source Data file.

parasites were cultured at 37 °C, 5% CO₂ in a humidified atmosphere. All cultures were regularly tested for *Mycoplasma*.

### Primary DCs and splenocytes

Murine bone marrow-derived DCs were generated as previously described[66]. Bone marrow cells were purified from 4–8 week-old mice and cultured in RPMI 1640 medium (VWR, Cat# 392-0427) supplemented with 10% FBS, 20 μg/ml gentamicin, 10 mM HEPES and 10 ng/ml recombinant GM-CSF (PeproTech, Cat# 1479-1407). DCs (loosely adherent cells) were harvested after 6 days. Spleens were isolated from 4–8 week-old mice and mashed using a 40 μm cell strainer. Splenocytes were collected and RBC lysed with RBC-lysis buffer (Invitrogen, Cat# 00-4300-54).

### Preparations of *T. gondii*-infected cells and extracellular tachyzoites

Inoculations with infected cells were performed as previously described[37,66]. Briefly, DCs or splenocytes were pre-labeled with CFSE (Invitrogen, Cat# C34554) or CMTMR (Invitrogen, Cat# C2927) and challenged with freshly egressed *T. gondii* tachyzoites at multiplicity of infection (MOI) 1 (type I, RH) or MOI 2 (type II, PRU, ME49; type III, CTG) for 4 h to obtain an infection frequency of ∼50%. CTG was pre-labeled with CFSE. Cell suspensions were then washed thrice and spun at 60–70 x g to minimize free tachyzoites in the inoculum. Total numbers of colony-forming units (cfu) injected into animals was confirmed by plaquing assays. Host cell number and viability was assessed by ocular hemocytometry and flow cytometry, respectively.

For inoculations with free tachyzoites, tachyzoites were collected from monolayers with maximum 50% HFF lysis, passed twice through a 27-gauge needle, washed twice by centrifugation at 870xg, kept at RT and inoculated in mice within 30–60 min from collection. Total numbers of cfu injected into animals was confirmed by plaquing assays.

### Intraperitoneal (ip) and intravenous (iv) inoculations in mice

Six to 8 week-old mice were inoculated ip with 0,2–10 × 10⁶ cfu of *T. gondii* tachyzoites. To assess parasite loads, animals were sacrificed at 2, 3 or 6 dpi. When indicated, anti-mo CD45 Alexa fluor 647-conjugated antibody (R&D Systems, Cat# FAB3507R, Clone 319211, Lot# 1726103) was injected iv (0,4 mg/kg) 10 min before euthanasia. For adoptive transfer assays, mice were inoculated iv with 20 × 10⁶ cfu of *T. gondi*-challenged DCs and sacrificed 24 hpi.

### Microsurgery, intracarotid artery inoculations and treatments

Six to 8-week-old mice were anesthetized with 2% isoflurane (Sigma-Aldrich, Cat# CDS019936) and the common carotid artery (CCA), external carotid artery (ECA) and internal carotid artery (ICA) were exposed under a dissecting microscope (Leica M205 FA). The left CCA was carefully separated from the vague nerve using micro-forceps. A 6–0 silk ligature was tied around the CCA and the ECA. A new loosely ligature was placed in the CCA below the bifurcation site. Using micro-scissors, a small cut in the CCA was made between the first and loose ligatures. A 32gauge/.8Fr catheter (Instech, Cat# BTPU-010) fitted on a syringe with 100 μl of medium containing cell/parasite suspensions was inserted into the CCA, guided to the bifurcation site and tied with a new ligature. The loose ligature was opened and the fluid was slowly injected over 5 min into the ICA. Then, the loose ligature was tightened and the catheter removed. Finally, fat tissue and muscles were replaced, the skin sutured and analgesic (Temgesic, Eumedica Pharmaceuticals) administrated. When indicated, 100 U/kg of heparin (ThermoFisher Scientific) was injected iv 2 h after ICA inoculation. Alternatively, 2 mg/kg of LPS (Sigma-Aldrich) was injected ip 4 h prior the surgical procedure. For blocking assays, anti-mo CD54 (eBioscience, Cat# 14-0541-85, Clone YN1/1.7.4, Lot# 2089516), anti-mo LFA-1 beta (eBioscience, Cat# 14-0181-85, Clone M18/2, Lot# 2577324) or isotype (eBioscience, Cat# 14-4321-85, Clone eBR2a, Lot# 2777082) (all 0,2 mg/kg) were mixed with the cell/parasite suspensions 5-10 min prior to ICA inoculation.

### Flow cytometry

DCs were challenged with freshly egressed tachyzoites for 5 h, washed in PBS and stained in FACS buffer (0,5% BSA, 2 mM EDTA in PBS). Challenged DCs were stained with 1/50 (v/v) PE-Cyanine7 anti-mo CD11c (eBioscience, Cat# 25-0114-82, Clone N418, Lot# 2142958), 1/50 (v/v) PE anti-mo CD18 (eBioscience, Cat# 101407, Clone M18/2. Lot# B313003) and 1/ 50 (v/v) Super Bright 702 anti-mo MHCII I-A/I-E (eBioscience, Cat# 67-5321-80, Clone M5/114.15.2, Lot# 2020037) antibodies and analyzed on a LSR Fortessa cell cytometer (BD Biosciences). Blood was collected by cardiac puncture into heparinized tubes and peripheral blood mononuclear cells (PBMCs) were purified with Lymphoprep™ (STEMCELL Technologies, Cat# 07801). Cells were

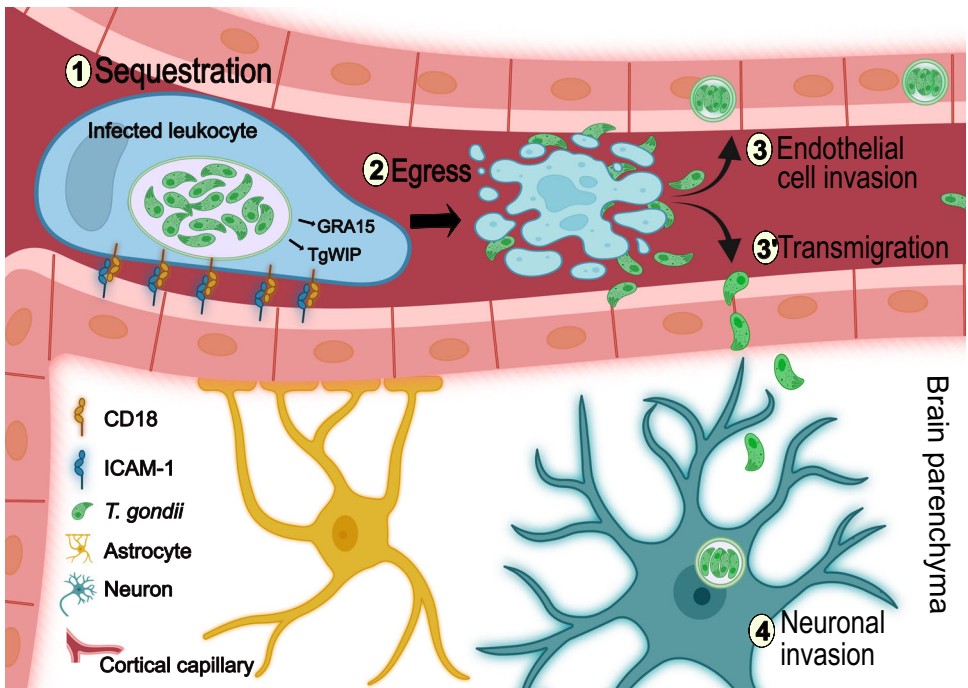

**Fig. 6 | Sequestration of *T. gondii*-infected leukocytes at the BBB facilitates CNS colonization.** The cartoon model illustrates the sequential processes involved in the CNS invasion by *T. gondii*, based on experimental data. **1** Initial adhesion: Circulating phagocytes infected with *T. gondii* adhere to the endothelium of cortical capillaries. This adhesion is mediated by ICAM-1/CD18 interactions, which facilitate the attachment of infected DCs. Microvascular inflammation, induced either by intraperitoneal infection with *T. gondii* or by LPS treatment, rapidly exacerbates the sequestration of infected leukocytes. In contrast, heparin treatment or treatments with blocking antibodies to ICAM-1 or CD18 reverse this sequestration. The *T. gondii* effectors TgWIP and GRA15, secreted into the cytosol of infected leukocytes, promote adhesion to the endothelium. **2** Intracellular replication and egress: After intracellular replication, tachyzoites egress from the sequestered leukocytes. In absence of parasite replication and egress, infected leukocytes remain sequestered for >48 h. **3** and **3'** Infection of endothelial cells or transmigration: Egressed

tachyzoites can either infect endothelial cells or directly transmigrate across the endothelium. **4** Infection of neurons: Upon transmigration, *T. gondii* rapidly infects cortical neurons. Vacuoles with intra-neurally replicating parasites were retrieved 16 h after inoculation in the cerebral circulation. Additionally, experimental data indicate two alternative routes of CNS invasion, which are not depicted in the cartoon. First, *T. gondii*-infected DCs can directly transmigrate into the parenchyma. Second, circulating extracellular tachyzoites can either infect endothelium or directly transmigrate to the parenchyma following intracarotid or intravenous inoculation, without replication in the endothelium. However, the frequency of these alternative routes appears significantly lower than the primary pathway early during infection, which involves sequestration of parasitized leukocytes and parasite egress, followed by either endothelial cell invasion or transmigration across the endothelium. Created in BioRender. Pairoto, M. (2025) https://BioRender.com/i14p970.

fixed in 4% PFA (Histolab, Cat# 02176), resuspended in FACS buffer and analyzed on LSR Fortessa. Data were analyzed with FlowJo software v10.

### Quantification of *T. gondii* foci in cortical brain sections

To visualize cerebral blood vessels, mice were injected iv with 50 µl of PBS 3% Evans blue (Sigma, Cat# E2129-106) and 5% BSA (Sigma-Aldrich) prior brain extraction. Brains were washed in PBS and fixed in 4% PFA for 48 h at 4 °C. Then, PFA was replaced by PBS 10% sucrose (Sigma-Aldrich) for 2 d at 4 °C. Fixed brains were frozen in liquid nitrogen and 50 µm thick cryosections were collected on glass slides. To quantify the number of infected DC/*T. gondii* foci, the prefrontal cortex area of the brain was imaged using epifluorescence microscopy (Leica DMi8). A *T. gondii* foci was defined as a single replicating vacuole or a localized group of tachyzoites that was not confined inside or associated with CMTMR⁺ DCs. Data were expressed as the number of *T. gondii* foci or infected DCs per cm² of tissue. Images were manually quantified with Fiji/ImageJ software. Data tables were exported and plotted using GraphPad Prism v.10 software.

### 3D Immunohistochemistry and image processing

Brain sections (50 µm thick) were transferred into 2 ml U-bottom tubes (5 sections/ tube) and washed 3 times with PBS. Then, samples were permeabilized and blocked in 1 ml of blocking buffer (1% Triton X-100

and 10% FBS in PBS) for 2 h at RT. Then, samples were incubated with 0,2 ml of primary antibodies anti-NeuN (Abcam, Cat# ab177487, Clone EPR12763, Lot# GR3275122-10) and anti-GPAF (Invitrogen, Cat# 13-0300, Clone 2.2B10, Lot# VH307339) diluted 1/250 (v/v) in blocking buffer for 48 h at 4 °C with gentle shaking. Sections were washed in PBS 0.1% Triton X-100 (3 times x 1 h) and incubated with 0,2 ml of Alexa Fluor 594-conjugated secondary antibodies (Invitrogen, Cat# A21442, Lot# 2906378; Molecular probes, Cat# A2147, Lot# 52521 A) diluted 1/1000 (v/v) in blocking buffer for 24 h at 4 °C with gentle shaking. After washing with PBS 0,1% Triton X-100, samples were mounted for imaging with LSM 800 Airyscan, Zeiss confocal microscope.

To quantify the localization of *T. gondii* in relation to blood vessels, DCs, neurons and astrocytes, brain sections were imaged using laser lines 488 nm (for GFP and CFSE), 561 nm (for RFP, CMTMR and Alexa 594) and 640 nm (For Evans blue and Alexa 647). Z-stacks from brain sections were collected with 40x objective from 0 to 50 µm in depth with 0.5-1 µm interval in the vertical z-axis and processed with IMARIS v.10.1 software. The *Surface* rendering tool was used to automatically define brain vasculature, DCs, *T. gondii*, neurons and astrocytes. The diameter (µm) of the vessels, distance to the branching points, and distance to the nearest vessel were manually defined and data tables were exported and plotted with GraphPad Prism v.10 software. Infected DCs/neurons/astrocytes were defined with the *Surface-Surface Colocalization* tool.

## RT-Quantitative Polymerase Chain Reaction (PCR)

Total genomic DNA was purified from brain, liver and lung using DNeasy blood & Tissue kit (Qiagen, Cat# 69506) according to manufacturer's protocols. Organs were homogenized in 4 ml of sterile PBS on 70 μm cell strainers. 0.5-1 ml of tissue homogenate was pelleted in 1.5 ml tubes for gDNA purification. gDNA concentration was measured using a spectrophotometer (Nano Drop 1000, ThermoFisher Scientific). Real-time PCR was performed using 100 ng gDNA, forward and reverse primers (200 nM) targeting *TgB1 gene* or *mGAPDH gene* and SYBR green PCR master mix (Kappa Biosystem, Cat# KK4602) with a QuantStudio™ 5 real-time PCR system (ThermoFisher). GAPDH was used as a house-keeping gene to generate ΔCt values in order to calculate relative expression ($2^{-\Delta\Delta CT}$).

For brain micro-vessels, total RNA was extracted using RNeasy mini kit (Qiagen, Cat# 74104) according manufacture's protocol. The RNA was quantified by spectrophotometry (NanoDrop 1000). cDNA was synthesized with Maxima first strand cDNA synthesis kit (ThermoFisher, Cat# EP0753). Real-time PCR as performed using 10–100 ng cDNA, 200 nM forward and reverse primers and SYBR green PCR master kit (Kapa Biosystem). Primers are listed in Table S1. GAPDH and HRT were used as house-keeping genes to generate ΔCt values in order to calculate relative expression ($2^{-\Delta\Delta CT}$).

## Isolation of cerebral micro-vessels

Brain micro-vessels were isolated as previously described[39]. Brains were homogenized by passage through a 23-gauge needle. Homogenates were diluted 1:1 (v/v) with 30% dextran solution (MW 70.000; Sigma-Aldrich, Cat# 31390-25) and centrifuged at 10.000xg for 15 min at 4 °C. The myelin layer was discarded and the pellet resuspended in DMEM 10% FBS. The suspension was passed through a 40-μm cell strainer and the vessels fragments retrieved were washed with PBS. Finally, the cell strainer was back-flushed with PBS and the micro-vessels pelleted by centrifugation at 4500 x g for 30 min at 4 °C.

## Plaquing and adhesion assays

For plaquing assays[66], brains, livers, and spleens from infected mice were extracted and homogenized using cell strainers (70 μm pore size). Homogenates were diluted in DMEM 10% FBS at 1/10 to 1/1000 (v/v) and added to confluent HFF monolayers in 24-well plates. After 24 h, medium was replaced. The numbers of viable parasites per g of tissue were determined by counting plaque formation after 3–5 days. To assess parasite viability, freshly egressed tachyzoites were resuspended in DMEM with 10% FBS (800 tachyzoites/ml), 250 μl were added to confluent HFF monolayers in 24-well plates at 0, 2, and 4 h after harvest and the number of viable parasites was determined by plaque formation.

For adhesion assays, DCs were pre-labeled with CMTMR or CFSE and challenged with freshly egressed GFP- or RFP-expressing tachyzoites (MOI 1 for type I, MOI 2 for type II). 1–5 × 10³ cfu of *T. gondii* challenged DCs were added to confluent bEnd.3 monolayers for 1 h. Then, monolayers were washed 5 times x 5 min with PBS under shaking (300 rpm), fixed 5 min in PFA 4%, washed with PBS and the number of infected DCs per well was quantified by epifluorescence microscopy. When indicated, monolayers were pre-treated with LPS (100 ng/ml) overnight. Before the adhesion assay, LPS-treated monolayers were washed 5 times with DMEM 10% FBS. Alternatively, heparin (100 U/ml) was mixed with the challenged DCs immediately before adding them to the monolayers. For blocking assays, anti-mo ICAM-1 or isotype (0,5 g/ ml) were mixed with the cell suspensions and immediately added to the monolayers.

## Statistical analyses

Statistical analyses were performed with GraphPad Prism v. 10. For multiple sample comparisons, one-way ANOVA followed Bonferroni's multiple comparison test were carried out on data sets with normal distribution, and Kruskal–Wallis test followed Dunn's post-hoc test was performed on data sets with non-normal distribution. For data sets from 2 samples with normal distribution, two-tailed Student's *t*-test was performed. For data sets from 2 samples with non-normal distribution, Mann–Whitney U-test was performed. A *p*-value < 0.05 was defined as significant differences for all statistical tests.

## Reporting summary

Further information on research design is available in the Nature Portfolio Reporting Summary linked to this article.

# Data availability

Source data are provided in this paper. All data needed to support the conclusions are presented in the paper, the Supplementary Information and the Source Data file. Source data are provided with this paper.

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

## Acknowledgements

We thank Drs. Jeroen Saeij, Ali Hakimi, Chris Hunter and David Bzik for parasite lines, Dr. Raad Askar for expert veterinary advice, the Imaging Facility Stockholm University (IFSU), the In Vivo Imaging Facility Stockholm University (IVIMSU) and the Experimental Core Facility (ECF). The studies were funded by the Swedish Research Council, grants 2018-02411 (A.B.), 2022-00520 (A.B.), The Swedish Brain Foundation (Hjärnfonden, grant FO2024-0022-HK-19 to A.B.), Åhlen Foudation grant 223020 (A.B.) and the Sven and Lily Lawski Foundation (M.E.R./A.B.).

## Author contributions

M.E.R. and A.B. developed the concept. M.E.R. developed method and performed microsurgery. M.E.R., A.H., N.L., F. H-O. and A.L.H. performed experiments, image processing and analyzed data. M.E.R. and A.B. wrote the manuscript.

## Funding

## Competing interests

The authors declare that no competing interests exist.
