## [Transparent Peer Review file · Nature Communications]

ICAM-1/CD18-mediated sequestration of parasitized phagocytes in cortical capillaries promotes neuronal colonization by *Toxoplasma gondii*

Corresponding Author: Professor Antonio Barragan

Version 0:

Reviewer comments:

Reviewer #1

(Remarks to the Author)

The infected peripheral mononuclear cells were first sequestered in cortical capillaries and after egressing, they rapidly infected cortical neurons.

The highlight that the microvascular inflammation due to the infection elevated the sequestration of parasitized immune cells and blocking the ICAM-1/CD18 leukocyte adhesion reverted sequestration.

Finally, the data indicate that TgWIP and GRA15, the secreted parasite effectors were involved in leukocyte hypermigration and inflammatory activation, and strain genotype-dependently elevated numbers of sequestered parasitized dendritic cells in capillaries and subsequently cerebral parasite loads.

How the parasite *Toxoplasma gondii* enters and colonizes the central nervous system has been discussed in the previous and current years by several research groups. The authors here used a novel experimental system and elegantly applied the parasitized phagocytes directly into the cerebral circulation.

This is a very interesting manuscript and the experiments are well designed. The mechanism by which which parasites colonizes the brain was not fully elucidated before, so this manuscript is filling the gap.

Only few recent references are missing for example this one DOI: 10.1186/s12974-021-02370-1

Reviewer #2

(Remarks to the Author)

The manuscript presented by Rodriguez et al. focuses on how *Toxoplasma gondii* crosses the blood-brain barrier (BBB) to reach the brain parenchyma and infect neurons. The work emphasizes the role of parasitized leukocytes, particularly dendritic cells (DCs), in facilitating this process. The authors designed a clever experimental approach to deliver *T. gondii*-infected cells into the carotid and thus directly into the cerebral circulation of mice to explore the mechanisms underlying BBB crossing and neuronal infection.

Main findings:

Sequestration in capillaries: The authors show that parasitized leukocytes, particularly DCs, get sequestered in cortical capillaries of injected mice. This sequestration has clear strain-specific elements, i.e., with Me49 and RH parasitized cells showing varying capacities to adhere to endothelial cells and capillary walls.

Parasite egress and neuronal infection: After sequestration, the parasite replicates within the DCs and subsequently egresses, invading neurons. This rapid translocation into neurons occurs without the need for prior infection or replication of endothelial cells. The timeline is for appearance of tachyzoites in the brain parenchyma is sufficiently short to support direct translocation upon egress from adherent DCs. This does not exclude that a translocation route via replication in endothelial cells is utilized in natural infections as demonstrated in previous work from this lab.

The role of ICAM-1/CD18 adhesion axis: Using blocking antibodies the authors provide direct proof that ICAM-1/CD18 interaction is crucial for the adhesion of parasitized DCs to the cerebral vasculature. Blocking this pathway significantly reduces the sequestration and overall parasite load in the brain.

Inflammatory exacerbation: External modulation of the inflammatory response, by administering ip LPS or iv heparin, dramatically enhances or decreases leukocyte adhesion and sequestration in capillaries, respectively. suggesting that

inflammation could exacerbate *T. gondii* brain infections.

Parasite effector molecules: *T. gondii* effector proteins TgWIP and GRA15 were found to modulate the migratory and adhesive properties of parasitized DCs, impacting the level of sequestration and overall cerebral parasite load.

Strengths:

Overall, the story is well organized and the case for sequestration-associated facilitation of tachyzoites crossing the bbb into the brain parenchyma of mice is carefully built. Importantly, the new findings presented here are in line with previous work and add to the robustness of the model as well as presenting new elements.

Innovative approach: The direct inoculation of parasitized cells into the cerebral circulation allows for a more accurate analysis of early infection events, overcoming the limitations of systemic infection models. While direct administration of DCs is still artificial the experimental setup allows for rigorous control of conditions and readouts. Thus, the experimental design is well suited to address the question about transmigration and sequestration.

Mechanistic insights: The study links the relevance of the host components mediating sequestration, i.e. ICAM-/CD18, with parasite factors GRA15 and TgWIP modulating adhesive capacities and motility of infected DCs.

Potential clinical relevance: While restricted to a controlled animal model the findings add another aspect increase our understanding of cerebral toxoplasmosis. This may be particularly relevant in immunocompromised patients where inflammation is often a complicating factor.

Limitations:

Limited in vivo inference: While the experimental system provides valuable insights, the direct injection of parasitized cells into the cerebral circulation may not fully replicate natural infection pathways. Despite revealing an additional route for translocation of tachyzoites across the bbb the model bypasses other organs and immune responses that would typically modulate infection.

Strain-specific findings: The results are highly dependent on the specific *T. gondii* strains used, which may limit the generalizability of the findings to other strains or natural infections. In particular the absence of functional Gra15 in RH would allow an additional experimental approach using ectopic expression of Me49 Gra15 in RH.

Possible inflammation-induced artefacts: The exacerbation of sequestration under inflammatory conditions was elicited by injecting LPS ip. Although the effect on the peripheral vasculature is evident, this experimental setup may not fully reflect the subtleties of immune responses in natural infections, raising questions about the translation of these findings to clinical settings. This is mitigated by the findings described in lines 163-170.

Overall, this reviewer finds no major issues with this carefully presented work. Few minor points are typos and missing/wrong words in several sentences which should be amended during the revision process. The manuscript provides a detailed exploration of *T. gondii*'s mechanisms for translocation across the bbb and subsequent neuronal infection, highlighting the role of leukocyte sequestration and parasite effector proteins. While the experimental model is highly controlled and informative, its limitations in replicating natural infection pathways should be weighed when interpreting the results.

Reviewer #3

(Remarks to the Author)

This paper continues the Baragan's lab interest in studying how *Toxoplasma* crosses the BBB and enters the brain. Here, they show that parasitized DCs become sequestered in cortical capillaries where the parasites egress and then translocate across the blood brain barrier. They demonstrate that ICAM1/Integrin Beta2 (CD11/CD18) interactions are important for this sequestration and that the *Toxoplasma* secreted effectors, GRA15 and TgWIP, also contribute to sequestration. The authors introduce a unique model of intracarotid delivery of free parasites as well as parasitized DCs to generate some of these data and insights. The data are interesting to the field but several issues need to be addressed.

1. The primary concern for this reviewer is that the majority of the data are predictable extensions of work published by this group. For example, the authors earlier showed that GRA15 induced ICAM1 was important for *Toxoplasma* transmigration across polarized epithelium and this work is largely the in vivo confirmation of those in vitro data. In this reviewer's opinion, this limited insight is not consistent with expectations for publication in Nature Communications.
2. What is novel is the finding that parasitized DC adhesion to cortical capillaries and transmigration into the brain were enhanced when mice were infected IP before intracarotid inoculation. This was unfortunately not followed up on.
3. Work by the Hunter lab suggested that *Toxoplasma* infect, replicate within endothelial cells, and then the endothelial cell lyse which allows the parasites to enter the brain. The discussion, at the very least should address differences between the two studies.
4. For this reviewer, the data here seem to refute the Barragan lab's earlier model suggesting that DCs are used as Trojan Horses for *Toxoplasma* to invade the brain. Is this correct? If so, this should be discussed in more detail.
5. The finding that type II strain parasites are better at entering the brain when they are within DCs may be misinterpreted. This is because free type II strain parasites are significantly less viable than when inside of an infected cell as well as less viable than free Type I strain parasites. Thus the data in Figs 3I,J could be misleading and interpreted differently than the authors have.
6. To my knowledge, the model in Figure 6 was misleading as it made it appear that neurons are in direct contact with endothelial cells to the same extent and astrocytic end feet are. This leads to the question of whether free parasites traffic to the neuron, use astrocytes as an intermediary, or induce neuronal (cell or dendrite) migration towards the BBB. The latter would support the model in Figure 6 but data are required for that model to be published as is.

Version 1:

Reviewer comments:

Reviewer #2

(Remarks to the Author)

The authors' rebuttal to the critique of their manuscript is very detailed and satisfies not only my concerns, but in my opinion also addresses those of the two other reviewers in great detail. Taken together with new data and additional experiments, I recommend publication of this work in Nature Communications.

Reviewer #3

(Remarks to the Author)

My concerns have been addressed.

Responses to Reviews - NCOMMS-24-57680A

Reviewer #1 (Remarks to the Author):

The infected peripheral mononuclear cells were first sequestered in cortical capillaries and after egressing, they rapidly infected cortical neurons.

The highlight that the microvascular inflammation due to the infection elevated the sequestration of parasitized immune cells and blocking the ICAM-1/CD18 leukocyte adhesion reverted sequestration.

Finally, the data indicate that TgWIP and GRA15, the secreted parasite effectors were involved in leukocyte hypermigration and inflammatory activation, and strain genotype-dependently elevated numbers of sequestered parasitized dendritic cells in capillaries and subsequently cerebral parasite loads.

How the parasite *Toxoplasma gondii* enters and colonizes the central nervous system has been discussed in the previous and current years by several research groups. The authors here used a novel experimental system and elegantly applied the parasitized phagocytes directly into the cerebral circulation.

This is a very interesting manuscript and the experiments are well designed. The mechanism by which which parasites colonizes the brain was not fully elucidated before, so this manuscript is filling the gap.

Only few recent references are missing for example this one DOI: 10.1186/s12974-021-02370-1

We appreciate the reviewer's evaluation of our study's experimental rigor, the novelty and the significance of our findings.

We agree that the suggested reference provides precision to the manuscript, especially regarding the inflammatory response. Figueiredo et al. includes analyses conducted as early as 3 days post-inoculation (3, 5, and 7 days post-inoculation), showing increased inflammation and immune cell presence around the choroid plexus endothelium. The study characterizes the early immune response at the choroid plexus and suggests that "choroid plexus infection is the initial step in the development of neuroinflammation." These findings align well with our observations too, which indicate early activation of the cerebral endothelium by days 3-5 (Olivera et al, *eLife*, 2021, fig 6). In the present manuscript, we present evidence of activation of the blood-brain barrier (BBB) microvasculature as early as 24, 48, and 72 hours post-infection with *T. gondii*, suggesting that inflammatory endothelial activation begins even earlier than previously documented at 3 days post-inoculation.

A comment and the reference have now been included in the discussion (page 9; lines 299-).

Figueiredo CA, Steffen J, Morton L, Arumugam S, Liesenfeld O, Deli MA, Kröger A, Schüler T, Dunay IR. **Immune response and pathogen invasion at the choroid plexus in the onset of cerebral toxoplasmosis.** *J Neuroinflammation*. 2022 Jan 13;19(1):17. doi: 10.1186/s12974-021-02370-1. PMID: 35027063; PMCID: PMC8759173.

Reviewer #2 (Remarks to the Author):

The manuscript presented by Rodriguez et al. focuses on how *Toxoplasma gondii* crosses the blood-brain barrier (BBB) to reach the brain parenchyma and infect neurons. The work emphasizes the role of parasitized leukocytes, particularly dendritic cells (DCs), in facilitating this process. The authors designed a clever experimental approach to deliver *T. gondii*-infected cells into the carotid and thus directly into the cerebral circulation of mice to explore the mechanisms underlying BBB crossing and neuronal infection.

Main findings:

Sequestration in capillaries: The authors show that parasitized leukocytes, particularly DCs, get sequestered in cortical capillaries of injected mice. This sequestration has clear strain-specific elements, i.e., with Me49 and RH parasitized cells showing varying capacities to adhere to endothelial cells and capillary walls.

Parasite egress and neuronal infection: After sequestration, the parasite replicates within the DCs and subsequently egresses, invading neurons. This rapid translocation into neurons occurs without the need for prior infection or replication of endothelial cells. The timeline is for appearance of tachyzoites in the brain parenchyma is sufficiently short to support direct translocation upon egress from adherent DCs. This does not exclude that a translocation route via replication in endothelial cells is utilized in natural infections as demonstrated in previous work from this lab.

The role of ICAM-1/CD18 adhesion axis: Using blocking antibodies the authors provide direct proof that ICAM-1/CD18 interaction is crucial for the adhesion of parasitized DCs to the cerebral vasculature. Blocking this pathway significantly reduces the sequestration and overall parasite load in the brain.

Inflammatory exacerbation: External modulation of the inflammatory response, by administering ip LPS or iv heparin, dramatically enhances or decreases leukocyte adhesion and sequestration in capillaries, respectively, suggesting that inflammation could exacerbate *T. gondii* brain infections.

Parasite effector molecules: *T. gondii* effector proteins TgWIP and GRA15 were found to modulate the migratory and adhesive properties of parasitized DCs, impacting the level of sequestration and overall cerebral parasite load.

Strengths:

Overall, the story is well organized and the case for sequestration-associated facilitation of tachyzoites crossing the bbb into the brain parenchyma of mice is carefully built.

Importantly, the new findings presented here are in line with previous work and add to the robustness of the model as well as presenting new elements.

Innovative approach: The direct inoculation of parasitized cells into the cerebral circulation allows for a more accurate analysis of early infection events, overcoming the limitations of systemic infection models. While direct administration of DCs is still artificial the experimental setup allows for rigorous control of conditions and readouts. Thus, the experimental design is well suited to address the question about transmigration and sequestration.

Mechanistic insights: The study links the relevance of the host components mediating sequestration, i.e. ICAM-/CD18, with parasite factors GRA15 and TgWIP modulating adhesive capacities and motility of infected DCs.

Potential clinical relevance: While restricted to a controlled animal model the findings add another aspect increase our understanding of cerebral toxoplasmosis. This may be particularly relevant in immunocompromised patients where inflammation is often a complicating factor.

We appreciate the reviewer's detailed evaluation of our study's experimental approach and the novel mechanistic insights that it provides. Indeed, the study of CNS invasion events by *T. gondii* early during infection has been limited. Studies have been partly hampered by the low parasite numbers reaching the brain parenchyma early during infection, forcing characterizations at later time points with exacerbated systemic infection. We have indicated this aspect in the introduction, results and revised discussion. The present approach by intracarotid (ICA) injection allowed obtaining quantitative data in a controlled fashion and at very early time- points.

Limitations:

Limited in vivo inference: While the experimental system provides valuable insights, the direct injection of parasitized cells into the cerebral circulation may not fully replicate natural infection pathways. Despite revealing an additional route for translocation of tachyzoites across the bbb the model bypasses other organs and immune responses that would typically modulate infection.

Indeed, we agree that every experimental approach has advantages and disadvantages and that often the combination of approaches complement each other and bring nuanced responses to the questions posed. We and others have previously characterized the invasion of the CNS using different inoculation routes, intraperitoneal and intravenous being the most commonly used (e.g., Konradt et al, *Nat Microbiol*, 2016; Olivera et al, *eLife*, 2021). The unexpected phenomenon of leukocyte sequestration occurred irrespective of inoculation route (see Fig. 1) but only intracarotid (ICA) delivery combined with confocal scanning of vast cortical areas (50 micrometer thick tissue sections) allowed obtaining quantitative data.

We agree with the pertinent observation by the reviewer that upon delivery in the ICA, immune responses that typically modulate infection are likely partly bypassed. For this reason, we performed pre-infection with *T. gondii*, before ICA injection. In the revision, we have expanded on this by performing pre-infection with both wild-type and a non-replicative strain of *T. gondii* (CPS), providing further characterization on the role of the early inflammatory response induced by *T. gondii*, including the upregulation of ICAM-1 in vascular endothelium (new Fig. S5E). Further, we show that treatment with anti-ICAM-1 reduces the cerebral parasite loads at 7 days post inoculation (new Fig. 5O, P). This, together with other results, show that the inflammatory response following pre-infection dramatically exacerbates sequestration. The effects of LPS already hinted to this result but we were surprised by the magnitude and rapidity that pre-infection impacted sequestration, with a very early inflammatory response of the microvasculature.

We have further clarified these aspects in Results (page 6) and the discussion (page 9-10).

Strain-specific findings: The results are highly dependent on the specific *T. gondii* strains used, which may limit the generalizability of the findings to other strains or natural

infections. In particular the absence of functional Gra15 in RH would allow an additional experimental approach using ectopic expression of Me49 Gra15 in RH.

We agree with the reviewer that generalizability to other strains is an important aspect. In the paper, we characterize a number of two prototypic lines of genotype I (RH, CPS) and two type II strains (PRU, ME49), broadly used in the field of research.

We have expanded our analysis to include a type III strain (CTG) and show that DCs infected by this strain also sequester. Thus, we can state the prototypic strains from the 3 predominant clonal lineages in humans and animals in Europe and North America (Sibley & Ajioka, *Ann Rev Microbiol*, 2008) can generate the phenomenon of sequestration. The new data has been added to new **Fig 1E-J**, Results (page 3.). We have also highlighted this in the abstract and in the discussion (page 7-8).

Although this is the first report on sequestration by *T. gondii*-infected leukocytes and extensive studies with multiple strains are likely needed for a global characterization, the data also indicate that genotype-related differences exist. In the new Fig. 1, the sequestration of type I, II and III-infected DCs is compared with measurable differences between strains. Later in the paper, we show that the expression of polymorphic effectors, which is known to differ between strains, e.g. GRA15 and TgWIP, impact sequestration. This is well in line with recent reports showing an impact on migratory phenotypes of infected phagocytes by polymorphic effectors (ten Hove et al, *Cell H&M*, 2022) and, further, the co-operative effects of polymorphic effectors (ten Hove et al, *mBio*, 2024). Probably, a similar reasoning could apply for other phenotypes, such as virulence in mice and strain-related immune responses (Sibley & Boothroyd, *Nature*, 1992; Mukhopadhyay et al, *Frontiers*, 2020).

However, we do not consider that GRA15 is essential for sequestration, as shown by the ability of the RH strain -that expresses a truncated or non-functional form of GRA15- to sequester and also the remnant sequestration by the GRA15 mutant. To address this, we have validated the specific role of GRA15 using a reconstituted line. The data shows that sequestration is restored in this line at comparable level to wild type, reinforcing the role of GRA15 (new **Fig S5D**).

Further, we argue in the revised discussion that co-operative effects between effectors GRA15 and TgWIP are likely in play here, in line with recent findings for migratory phenotypes/chemotaxis of infected phagocytes (ten Hove et al, *mBio* 2024). “Jointly, because the type I strain RH expresses a non-functional form of GRA15⁴⁹, TgWIP deficiency in type I RH may phenotypically reflect a combined GRA15/TgWIP deficiency in this respect. This likely explains the overall inferior sequestration frequency of Δ TgWIP-infected DCs compared with Δ GRA15-infected DCs at very early time points (1 hpi), that is, before parasite egress or significant replication sets in.” (page 8).

Altogether, the new data generalize the findings (type I, II and III) and reinforce the modulatory role of GRA15. The data have been included in the results section (new Fig 1, new Fig S5D). We have also highlighted this important aspect in the abstract (line 19) and revised discussion (page 6, lines 199-201).

Possible inflammation-induced artefacts: The exacerbation of sequestration under

inflammatory conditions was elicited by injecting LPS ip. Although the effect on the peripheral vasculature is evident, this experimental setup may not fully reflect the subtleties of immune responses in natural infections, raising questions about the translation of these findings to clinical settings. This is mitigated by the findings described in lines 163-170.

Yes, we agree with the reviewer. LPS and heparin are used in the paper as pro-inflammatory and anti-inflammatory/anti-adhesive compounds, respectively. As indicated by the reviewer, pre-infection with *T. gondii* is a more relevant immune-activating stimulus and yielded similar or superior effects compared with LPS.

As part of response to a question raised by Reviewer 3, we have added experiments by a non-replicating uracil-auxotroph *T. gondii* line (CPS) and report that single dose ip is also sufficient to induce measurable inflammation of the cerebral microvasculature within 24-48h (new Fig S5D), with an impact on sequestration (new Fig 5O, P). This reinforces the role of an early *T. gondii*-induced inflammatory response in sequestration.

Finally, we fully agree with the reviewer that this raises questions regarding clinical settings, and in extension, the remarkably stealthy colonization of the CNS by *T. gondii*. We have expanded on this important aspect in the revised discussion, exemplifying/contrasting with bacterial meningitis (page 9, lines 308-313)

Overall, this reviewer finds no major issues with this carefully presented work. Few minor points are typos and missing/wrong words in several sentences which should be amended during the revision process. The manuscript provides a detailed exploration of *T. gondii*'s mechanisms for translocation across the bbb and subsequent neuronal infection, highlighting the role of leukocyte sequestration and parasite effector proteins. While the experimental model is highly controlled and informative, its limitations in replicating natural infection pathways should be weighed when interpreting the results.

We appreciate the summary by the reviewer. Detected typos have been corrected and missing words added in the manuscript. In the revised discussion, we have weighed in limitations in replicating natural infections and expanded on the interpretation of the results, including clinical perspectives.

Reviewer #3 (Remarks to the Author):

This paper continues the Baragan's lab interest in studying how *Toxoplasma* crosses the BBB and enters the brain. Here, they show that parasitized DCs become sequestered in cortical capillaries where the parasites egress and then translocate across the blood brain barrier. They demonstrate that ICAM1/Integrin Beta2 (CD11/CD18) interactions are important for this sequestration and that the *Toxoplasma* secreted effectors, GRA15 and TgWIP, also contribute to sequestration. The authors introduce a unique model of intracarotid delivery of free parasites as well as parasitized DCs to generate some of these data and insights. The data are interesting to the field but several issues need to be addressed.

We thank the reviewer for the evaluation and we acknowledge the appreciation of the novel model. Point-by-point responses to address each issue are provided below.

1. The primary concern for this reviewer is that the majority of the data are predictable extensions of work published by this group. For example, the authors earlier showed that GRA15 induced ICAM1 was important for *Toxoplasma* transmigration across polarized epithelium and this work is largely the in vivo confirmation of those in vitro data. In this reviewer's opinion, this limited insight is not consistent with expectations for publication in Nature Communications.

We have carefully considered the perspective of the reviewer and share here our own perspective.

In this paper, we present, for the first time, evidence of (1) sequestration of parasitized leukocytes within cerebral cortical capillaries during *T. gondii* infection. Our findings reveal that this leukocyte sequestration promotes (2) rapid neuronal invasion, significantly impacting cerebral parasite loads. We demonstrate that (3) the ICAM-1/CD18 axis plays a crucial role in driving sequestration, and we identify (4) two parasite effectors, TgWIP and GRA15, that further enhance this process. Additionally, within 24 hours of intraperitoneal infection, we observed (5) a rapid inflammatory response and activation of the cerebral endothelium that exacerbates sequestration. To investigate this phenomenon, we established (6) a novel experimental model that involved microsurgical delivery of parasites and infected leukocytes directly into the cerebral circulation via the internal carotid artery. While leukocyte sequestration was observed with other inoculation routes (ip, iv), only the established experimental approach of internal carotid artery (ICA) injection, combined with high-resolution analysis of extensive cortical areas, consistently yielded quantitative data. Further, we perform a (7) comparison of dissemination by free tachyzoites and parasitized leukocytes for type I, II and III genotypes.

The original finding of leukocyte sequestration connects naturally to established concepts, such as the central ICAM-1/CD18 axis in leukocyte adhesion, and the role of parasite effectors TgWIP and GRA15, which have been previously associated with inflammation and modulation of leukocyte functions in other experimental contexts. These connections are crucial, as they contextualize and reinforce the central finding of sequestration.

We recently reported a role for ICAM-1 and GRA15 in the transmigration of infected DCs across endothelium in vitro (Ross et al., *Cell Molec Life Sci*, 2022). This previous work is referenced in the Results section and discussed in two sections of the Discussion. Here, we report, for the first time, a novel role for GRA15 in promoting cortical microvascular sequestration of infected DCs in mice. In fact, while the *predictable* result in vivo would have been enhanced transmigration (by extension of our in vitro findings), we instead found the -for us- *unpredictable* result of sequestration. Moreover, the finding that sequestered DCs remain sequestered in cortical capillaries for at least 48 h when infected with the non-replicating parasite CPS, indicates that sequestration and transmigration are not automatically linked (but may share common ligands mediating cell adhesion). When infected with replicating parasites, sequestered DCs lyse eventually, the tissue is invaded by egressing tachyzoites and neuronal invasion takes place, as shown in the paper for the first time.

This discovery of leukocyte sequestration offers a novel mechanism by which *T. gondii* can penetrate the brain parenchyma, potentially addressing several unanswered questions in the field (see response to Q4 for further detail). Overall, this manuscript provides the first description of the sequestration phenomenon, along with cellular characterizations and molecular identifications of host and parasite components involved.

Given the reasons outlined above, we believe that the concern that the majority of the data are predictable extensions is in discordance with the actual content of the manuscript. Further, the connections to ICAM-1 and GRA15 are important, though not central, as they contextualize and reinforce the central finding of sequestration, and provide molecular characterizations for the first time. However, we see the point that the reviewer wants to make that GRA15 and ICAM-1 have been previously associated to *in vitro* phenotypes in cellular models. We have clarified the discussion section regarding GRA15 and ICAM-1 by emphasizing the differences between previously reported *in vitro* results and our novel findings on sequestration *in vivo*. Additionally, we have highlighted the need for caution when extrapolating *in vitro* data to *in vivo* conditions. (page 7, lines 242-).

2. What is novel is the finding that parasitized DC adhesion to cortical capillaries and transmigration into the brain were enhanced when mice were infected IP before intracarotid inoculation. This was unfortunately not followed up on.

We agree this is an interesting aspect because it links the observed rapid inflammatory response of the microvasculature to the phenomenon of sequestration. We have followed up with additional experimentation upon pre-ip infection.

Pre-existing data in the manuscript with pre-ip approach addressed: rapid inflammatory response by microvasculature (Fig 4J-L, Fig S4), the effects of treatments with anti-ICAM-1 and anti-CD18 (Fig. 4F-P), the impact of GRA15 and TgWIP upon pre-ip (Fig. 5F-G). In addition, pro-inflammatory activation with LPS induced similar effect (Fig 5M, N).

Because ICAM-1 is elevated upon pre-ip and reduces sequestration, we have tested the effects of sequestration and de-sequestration by anti-ICAM-1 treatments on cerebral parasite loads beyond 1-48 h by assessing loads 7 days post-inoculation.

First, since pre-ip with wild-type *T. gondii* could likely contribute to the total cerebral parasite loads at this later stage, we instead used the non-replicative CPS strain. Prior to this, we confirmed that CPS induced an inflammatory response in the cerebral endothelium, similar to wild-type *T. gondii* (new **Fig. S5F, G**).

Next, a low dose of infected DCs (1×10^5 DCs, 5×10^4 cfu Tg type II) was then administered into the ICA \pm anti-ICAM-1 treatment (single dose) and parasite loads were assessed 7 days post-inoculation. Treated animals exhibited lower cerebral parasite loads compared to isotype control-treated mice, whereas splenic parasite loads remained unchanged (new **Fig. 5O, P**). Together, this shows that pre-ip induced inflammation elevates ICAM-1-dependent DC sequestration in the brain vasculature, with an impact on cerebral parasite loads.

These data have been added to the Results (page 6) and are brought up in the discussion (page 7, lines 238-242, 244-252, page 8-9, lines 324-336).

3. Work by the Hunter lab suggested that *Toxoplasma* infect, replicate within endothelial cells, and then the endothelial cell lyse which allows the parasites to enter the brain. The discussion, at the very least should address differences between the two studies.

Our original submission cites the paper by Konradt et al. a total of eight times throughout the manuscript. Specifically, four citations appear in various sections of the discussion, three in the Results section (related to differences in the experimental approaches), and one in the introduction. We believe that this distribution of citations (1+3+4) provides readers both with background and with ample opportunities to compare our study with Konradt et al.'s work.

We have included a table (Appendix 1) summarizing the main differences between the two studies. These differences have also been clarified in the results (page 3, line 90-, page 4, line 113; page 4, line 137) and discussion section (page 8, line 280-; page 9, lines 295-, page 9, line 324-, page 10, line 329-). Briefly, we believe the two studies complement each other, as they address different aspects of the topic, and we do not observe any directly contradictory results. In a previous paper (Olivera et al., *eLife*, 2021), we characterized endothelial invasion of tachyzoites using an intravenous (iv) approach similar to that of Konradt et al. However, as stated in the manuscript, only the experimental approach involving ICA injection enabled the quantification of sequestration.

Importantly, we have also contextualized our findings by drawing relevant comparisons with experimental models of various conditions, including: cerebral malaria, meningococcal meningitis, metastatic cancer, septicemia, experimental treatments for autoimmune encephalomyelitis and glioblastoma. These comparisons, along with their corresponding references, are presented on pages 8 and 9 of our manuscript. We believe these contextual discussions are highly relevant for the broad readership of the journal and enhance the overall impact of our study.

4. For this reviewer, the data here seem to refute the Barragan lab's earlier model suggesting that DCs are used as Trojan Horses for *Toxoplasma* to invade the brain. Is this correct? If so, this should be discussed in more detail.

We thank the reviewer for raising this point and agree that it warrants a nuanced response. We have taken this opportunity to provide further clarification on this specific point in our paper.

While our laboratory has indeed studied *T. gondii* dissemination from various angles - particularly focusing on the roles of infected DCs and other leukocytes in the parasite's systemic spread- the mechanisms by which *T. gondii* crosses into the CNS remain incompletely understood, both by our group and others in the field. Currently, there is no consensus among researchers regarding the precise pathway(s) *T. gondii* uses to access the CNS (Matta, Sibley et al., *Nat Rev Microbiol*, 2021).

Previous studies, including our own work and independent work from other laboratories, have demonstrated that parasitized phagocytes (monocytes, DCs) transport *T. gondii* to the brain, leading to elevated total parasite loads within this organ (e.g., Lambert et al., *Cell*

Microbiol, 2006; Courret et al., *Blood*, 2006). To date, we have not obtained any subsequent data, nor are we aware of any published findings, that contradict this conclusion. In fact, recent studies have extended this concept to include macrophages and further confirmed the role of monocytes and DCs in facilitating *T. gondii* dissemination to peripheral organs (e.g., Ten Hove et al., *Cell Host Microbe*, 2022; *mBio*, 2024).

A separate question remains regarding the specific role of infected DCs and other infected leukocytes in transporting *T. gondii* across the BBB and into the brain parenchyma.

Our lab has not previously reported experimental research on infected leukocytes crossing the BBB *in vivo* until the current study. Contrasting with the reviewer's statement, in a recent publication (Ross, Olivera, Barragan, *Trends Parasitol*, 2022), we state: "**direct translocation of parasitized leukocytes (Trojan horse) has been proposed [Lachenmeier, Liesenfeld et al, *J Neuroimmunol*, 2011] but thus far only shown *in vitro* across polarized primary brain endothelium [Ross et al, *CMLS*, 2021]**" and "**there is to date no evidence that translocation of parasitized DCs or other leukocytes across the BBB strictly promotes parasite colonization of the brain parenchyma**" and "**However, whether infected leukocytes deliver tachyzoites to the vasculature or traffic to the parenchyma remains unresolved.**". Similarly, in Schlüter & Barragan (*Frontiers Immunol*, 2019), we state: "**Hypermigration of infected DCs potentiates systemic parasite dissemination in mice, including the CNS, and may cooperate with chemotactic responses of DCs (Fuks et al, *PLoS Pathog*, 2012; Weidner et al, *Cell Microbiol*, 2013). However, hypermigration has not been linked specifically to passage across the BBB *in vivo* as discussed below (Kanatani et al, *PLoS Pathog*, 2017).**" Maybe this same argumentation is even more explicitly spelled out in (Fuks et al, *PLoS Pathog*, 2012): "**The present study does not address the passage of the parasite across the blood-brain barrier and whether the infected DC directly transport parasites into the CNS. Alternative but not mutually exclusive hypotheses are possible for the observed differences in parasitic loads in the brain.**".

Thus, our lab's perspective has consistently over time emphasized the importance of distinguishing between **(1)** the role of infected leukocytes for systemic dissemination of *T. gondii* to various organs, including transport to the brain *-organ-*, and **(2)** their putative ability to cross the BBB and deliver *T. gondii* to the parenchyma. In fact, this exact distinction served as the foundation for the experimental setup that led to the current findings. Making this nuance (1 vs 2) has proven to be pertinent given the novel findings in the current manuscript.

To date, a role for the direct passage of infected DCs or other infected leukocytes across the BBB has not been convincingly experimentally proven or disproven *in vivo*, to our knowledge. However, several independent studies have demonstrated the passage of infected leukocytes across *in vitro* BBB models and blood-retina barrier models (and similarly for intestinal barrier and placenta models *in vitro*). The "Trojan horse" pathway has been widely discussed in publicly available literature, including reviews and original publications. While it is sometimes endorsed and other times rejected, discussions are occasionally accompanied by speculative extrapolations about *in vivo* processes. In our opinion, it is likely prudent to approach such speculative assertions with caution and avoid endorsing or rejecting them without substantial evidence.

The focus of the current paper is on the discovery of the phenomenon of sequestration. However, we also test and compare with other possible pathways mediating CNS access. As shown in the current paper, we provide evidence that infected pre-labeled DCs can directly access the brain parenchyma. However, this passage occurs at a much lower frequency compared to the number of tachyzoites that reach the parenchyma after egress from sequestered DCs. A recent study by the Lodoen/Hunter labs (Schneider et al., *mBio*, 2022) also suggests that highly motile cells -presumably leukocytes- can transport parasite-containing vacuoles across the BBB, which aligns with our current findings on infected DCs.

Consequently, the data indicates that infected DC/phagocyte sequestration with subsequent parasite egress and invasion across the BBB is a more prevalent mechanism for *T. gondii* to enter the brain parenchyma compared to direct transport by infected DCs.

Most importantly, the current paper establishes a direct link between systemic dissemination in leukocytes and neuronal invasion, thereby reconciling these two phases of the infection for this obligate intracellular parasite through the phenomenon of sequestration.

These clarifications are now summarized in the discussion for the reader's benefit (pages 9-10, lines 324-339).

5. The finding that type II strain parasites are better at entering the brain when they are within DCs may be misinterpreted. This is because free type II strain parasites are significantly less viable than when inside of an infected cell as well as less viable than free Type I strain parasites. Thus the data in Figs 3I,J could be misleading and interpreted differently than the authors have.

We agree with the reviewer that parasites are likely “protected” inside the DCs and other infected leukocytes, as indicated by their replication in vacuoles inside DCs sequestered in mouse microvasculature. Earlier characterizations have shown that parasites have high viability in DCs in vitro and upon adoptive transfer in vivo (Lambert et al, *Infect Immun*, 2009; Fuks et al, *PLoS Path*, 2012). In contrast, extracellular parasites in the blood are likely exposed to complement and IgM, with neutralizing effects (Couper et al, *Infect immun*, 2005). In addition, for inoculations in mice, there might be practical experimental aspects that impact (for example, time from harvesting to injection) and inherent differences between type I and II strains (for example, gliding motility and invasiveness) (Barragan & Sibley, *J Exp Med*, 2001).

Regarding extracellular tachyzoites, the reviewer is likely referring to the observation that the viability of type II tachyzoites (ME49 or PRU strains) declines more rapidly than that of type I tachyzoites (RH strain) when exposed to extracellular conditions for extended periods. In our experience, these viability differences are significantly influenced by several factors: 1) parasite culture conditions, particularly the point of host cell lysis and synchronicity, 2) harvesting methods, and 3) total duration of extracellular exposure after tachyzoite collection (Barragan & Sibley, *J Exp Med*, 2001). In our experiments, we took measures to minimize the time between tachyzoite collection and mouse inoculation (see Methods, page 11, line 397). Additionally, we carefully monitored and controlled the viability of tachyzoites throughout the procedure.

The reviewer is correct that in the practical experimental handling of type I and type II parasites, there are measurable differences in viability. See for example, comparison of viability by plaquing assays between RH (~90% viability) and ME49-PTG (~70% viability) in Barragan & Sibley, *J Exp Med*, 2001. Similarly, in our viability control experiments, we see a difference between type I (RH) and type II (PRU, ME49). This difference is in the order of 20-25% (1.25 fold). We now provide viability data determined by plaquing assays comparing type I and II parasites under the conditions applied (new **Fig S3D**).

To address this issue, we conducted control experiments to compare parasite loads at different doses of type I and type II parasites. Our results indicate that achieving comparable parasite loads requires inoculating type II parasites at a dose four times higher (for tachyzoite counts ~400%, 4-fold; for cfu ~300%, 3-fold) than that of type I parasites (new **Fig S3C**). Thus, although the 20% viability difference between extracellular type I and type II may contribute, it cannot solely explain the manifold difference in parasite loads hours after inoculation. Similarly for type II-infected DCs (assuming 80-90% viability) and extracellular type II (assuming 50-60% viability) likely cannot explain the manifold (10-fold) difference in cerebral parasite loads 24 h after inoculation. However, we agree that tachyzoite viability must be considered a contributing factor.

We have added viability data to Results (page 4-5, lines 142-145, new **Fig S3B-D**) and brought up this aspect in the revised discussion (specifically in page 8, lines 291-292; with related general discussion in page 7, lines 226-227 and page 8, lines 283-289).

6. To my knowledge, the model in Figure 6 was misleading as it made it appear that neurons are in direct contact with endothelial cells to the same extent and astrocytic end feet are. This leads to the question of whether free parasites traffic to the neuron, use astrocytes as an intermediary, or induce neuronal (cell or dendrite) migration towards the BBB. The latter would support the model in Figure 6 but data are required for that model to be published as is.

We agree that the dendrite of the neuron cartoon was drawn too near the endothelium, possibly giving the appearance that neurons could go beyond astrocytes. There is no such claim in the manuscript and this has been corrected in the cartoon. The central aspect of this model is the sequestration of leukocytes and subsequent invasion of neurons by egressing tachyzoites, backed up by experimental data in the paper.

Precision on distances and cellular localizations at the BBB is also provided in the introduction (page 2, lines 39-42) and the discussion (page 8, lines 275-277). "The BBB (NVU wall) is < 10 μm thick, with an average cortical inter-capillary distance of 40 μm ^{7,8} and an average distance of 15 μm between neuronal somata and capillaries⁵⁴.", in line with the neuronal and vascular tissue stainings in Fig. 3.

The cartoon has been modified accordingly. We thank the reviewer for the thorough scrutiny of the cartoon.

In conclusion, we sincerely appreciate the three reviewers' valuable critiques and suggestions. We hope they will agree that the inclusion of new data and the accompanying discussion have significantly strengthened the manuscript. We believe these substantial revisions further support our original conclusions, and we respectfully request reconsideration of our manuscript for publication in *Nature Communications*.

Appendix1. Features and parameters studied in the indicated studies

Study	Central finding	Infection routes	Infection modes	Parasite genotype tested	Most studied timepoints	Neuron invasion kinetics	Host ligands	Parasite effectors	Cerebral microvessel inflammation
Konradt et al.	Infection of endothelium	Iv (ip, po)	-Free tachyzoites	Type I	7-9 days post-inoculation	ND	ND	ND	ND
Rodriguez et al.	Sequestration of infected leukocytes	ICA (ip, iv)	-Infected leukocytes /DCs/PBMCs/ splenocytes -Free tachyzoites	Type I Type II Type III	1-2 days post-inoculation	Yes (1 h – 72 h)	ICAM-1 CD18	TgWIP GRA15	-Days 1-3 -effects of T. gondii , LPS and heparin

ND: not determined